**Decision Support System version 1.0 (DSS v1.0) for air quality management in Delhi, India.**

Gaurav Govardhan[1,2,*], Sachin D. Ghude[1,*], Rajesh Kumar[3], Sumit Sharma[4], Preeti Gunwani[5], Chinmay Jena[5], Prafull Yadav[1], Shubhangi Ingle[1], Sreyashi Debnath[1,6], Pooja Pawar[1], Prodip Acharja[1,6], Rajmal Jat[1], Gayatry Kalita[1], Rupal Ambulkar[1], Santosh Kulkarni[7], Akshara Kaginalkar[7], Vijay K Soni[5], Ravi S. Nanjundiah[8,9], and Madhavan Rajeevan[10]

1: Indian Institute of Tropical Meteorology, Ministry of Earth Sciences, Pune, Maharashtra, India
2: National Centre for Medium-Range Weather Forecasting, Ministry of Earth Sciences, Noida, Uttar Pradesh, India
3: National Center for Atmospheric Research, Boulder, CO, United States of America
4: The Energy and Resources Institute, Delhi, India
5: India Meteorology Department, Ministry of Earth Sciences, Delhi, India
6: Savitribai Phule Pune University, Pune, Maharashtra, India
7: Centre for Development of Advanced Computing, Pune, Maharashtra, India
8: Centre for Atmospheric and Oceanic Sciences, Indian Institute of Science, Bengaluru, India.
9: Divecha Centre for Climate Change, Indian Institute of Science, Bengaluru, India.
10: National Centre for Earth Science Studies, Thiruvananthapuram, Kerala, India.
* : corresponding authors:
 Gaurav Govardhan: gaurav.govardhan@tropmet.res.in ,
Sachin D. Ghude: sachinghude@tropmet.res.in

**Abstract**

This paper discusses the newly developed Decision Support System version 1.0 (DSS v1.0) for air quality management activities in Delhi, India. In addition to standard air quality forecasts, DSS provides the contribution of Delhi, its surrounding districts, and stubble-burning fires in the neighboring states of Punjab and Haryana to the $PM_{2.5}$ load in Delhi. DSS also quantifies the effects of local and neighborhood emission-source-level interventions on the pollution load in Delhi. The DSS-simulated Air Quality Index for the post-monsoon and winter seasons of 2021-22 shows high accuracy (up to 80%) and a very low false alarm ratio (~20%) from Day 1 to Day 5 of the forecasts, especially when the ambient AQI is > 300. During the post-monsoon season (winter season), emissions from Delhi, the rest of the National Capital Region's districts, biomass-burning activities, and all other remaining regions on average contribute 34.4% (33.4%), 31% (40.2%), 7.3% (0.1%), and 27.3% (26.4%), respectively, to $PM_{2.5}$ load in Delhi. During peak pollution events (stubble-burning periods), however, the contribution from sources within Delhi (farm fires in Punjab-Haryana) could reach 65% (69%). According to DSS, a 20% (40%) reduction in anthropogenic emissions across all NCR districts would result in a 12% (24%) reduction in $PM_{2.5}$ in Delhi on a seasonal mean basis. DSS is a critical tool for policymakers because it provides such information daily through a single simulation with a plethora of emission reduction scenarios.

## 1. Introduction

The national capital of India, Delhi, is one of the most populated capitals in the world with an estimated count of more than 18.7 million (UIDAI, 2021). Immense population density, urbanization, and

industrialization within the city have resulted in many urban issues, including air pollution (Molina and Molina, 2004; Chopra, 2016; Zhang et al., 2022). The primary sources of pollutants are vehicles, industries, power plants, waste-burning practices, construction and demolition activities, road dust, etc. On top of this, the post-monsoonal (October-November) harvesting of the paddy crops and the associated burning of the paddy residue in the neighboring states of Haryana and Punjab also contribute to the degradation of air quality in Delhi and the surrounding region (Bikkina et al., 2019; Bray et al., 2019; Choudhury et al., 2019; Kulkarni et al., 2020; Nair et al., 2020). Besides, the geographical location and the local meteorological conditions, especially during the winter months, aggravate the pollution levels in the city (Guttikunda and Gurjar, 2012; Tiwari et al., 2014; Kumar et al., 2020). The pollution in the city is at its peak during the post-monsoon and the winter seasons, though the summer (April-June) months also bring severe dust storms and the associated degradation of Delhi's air quality (Banerjee et al., 2021; Chakravarty et al., 2021; Parde et al., 2022). The air quality in Delhi is so poor that it occasionally (especially during the post-monsoon and winter seasons) crosses the national air quality standards by more than ten times (Kanawade et al., 2020; Jena et al., 2021; Roozitalab et al., 2021). Owing to the ever-increasing pollution, Delhi has been topping the list of the most polluted national capital cities in the world (Meteosim, 2019). It has been estimated that the air pollution in Delhi is causing more than 7,000 premature mortalities every year (Guttikunda and Goel, 2013; Ghude et al., 2016; Saini and Sharma, 2020). The loss of average life expectancy in the city is also estimated to be around two years in Delhi (Ghude et al., 2016; Guo et al., 2018).

The primary solution to this problem lies in the reduction of anthropogenic emissions happening in and around the city. However, permanent mitigation of emissions is a long-term objective due to the involvement of multiple socio-economic factors (Riahi et al., 2017). A short-term and effective solution to this problem could be related to creating awareness in the common public about air pollution, releasing early warnings about the air pollution episodes that are likely to happen, and imposing temporary emission controls so that the exposure of the common people to acute levels of air pollution could be avoided. With this motivation, the Government of India, in the year 2018, directed the Ministry of Earth Sciences (MoES) to develop an early warning system for air pollution events happening in Delhi. With this mandate, the Indian Institute of Tropical Meteorology (IITM), Pune, and the India Meteorology Department (IMD) developed the 'Air Quality Early Warning System' (AQEWS) in collaboration with the National Center for Atmospheric Research (NCAR), USA, in 2018. AQEWS is a dynamical modeling system that simulates air quality over the entire India with a special focus on Delhi (Ghude et al., 2020; Kumar et al., 2020; Jena et al., 2021; Sengupta et al., 2022). The forecasting for Delhi is carried out with a spatial grid spacing of 400 m x 400 m. The system is capable of delivering forecasts for three days and at a slightly coarser resolution (10 km) for the next ten days. The skill of these forecasts has been found to be excellent, especially when the air quality is beyond the 'very-poor' category (Jena et al., 2021; Sengupta et al., 2022). The forecast has been found to be very useful to policymakers and has helped them manage the air quality in the city, especially when severe air pollution episodes are predicted (Ghude et al., 2022).

However, the governing authorities require more specific information about the emission sources contributing to forthcoming air pollution events occurring in the near future besides the actual forecasts. They also want to know the solution on how to reduce the impact of an air pollution event forecasted to affect the city. These requirements were put forth by the Commission for Air Quality Management (CAQM) in the National Capital Region and Adjoining Areas, constituted by the honorable Supreme Court of India in 2021. While there exist some recent source-apportionment-related studies on air pollution in Delhi (e.g., Gadi et al., 2019; Guo et al., 2019; Shivani et al., 2019; Rai et al., 2020; Tobler et al., 2020; Yadav et al., 2020; Hama et al., 2021; Lalchandani et al., 2021), there does not exist a system that can provide source apportionment information about the city's pollution either in near-real-time or 72 h in advance. Even globally, a very few such systems exist (Denby et al., 2020; Colette et al., 2022) which give real-time and forecast of region-wise source apportionment of air pollution. Such a capability

is highly essential to suggest possible short-term immediate-relief-based solutions to the pollution menace happening in Delhi, especially during the post-monsoon and winter seasons. Responding to this requirement from the CAQM, we have come up with a dynamical modeling system named 'Decision Support System' (DSS) for air quality management in Delhi. The DSS is a new armor in our AQEWS that has already been providing neighborhood scale forecasts in Delhi (Jena et al., 2021) and provides quantitative information about the

a) the contribution of emissions from 20 districts of the National Capital Region (NCR) (including Delhi) to the air pollution ($PM_{2.5}$ and CO) in Delhi,

b) the contribution of eight different emission sectors within Delhi to the air pollution in the city,

c) the contribution of emissions from the biomass-burning activities happening in the neighboring states of Punjab and Haryana to the degradation of air quality in Delhi, and

d) the efficacy of the possible emission source-level interventions on the forecasted air pollution event occuring in Delhi.

The DSS was operationalized during the post-monsoon and the winter seasons of the year 2021. It has been found to be very helpful for the governing authorities and the policy-makers. It has been estimated that the governing authorities avoided a severe air pollution event in Delhi by improving the air quality index (AQI) in the city by 20-22%, taking guidelines from the AQEWS and DSS (Ghude et al., 2022). Keeping in mind the usefulness of DSS, the CAQM has recommended that DSS must be an integral part of the decision-making process for reducing air pollution in the NCR (CAQM, 2022).

In this paper, we describe DSS by explaining its underlying modeling system, the various input datasets needed for the simulations, and the chemical data assimilation occurring in the system, in section 2. In the results section (section 3), we first evaluate the performance of DSS in capturing air pollution load in Delhi during the post-monsoon and the winter seasons of the year 2021-22. This is followed by the source-apportionment-related results from DSS for both the seasons of interest. We further discuss the findings from the 'scenarios of emission reductions' from DSS. In section 4, we summarize the main results from the paper.

## 2. Details of the Modeling System

### 2.1 Domain and Meteorological Formulation

The DSS holds the fully coupled regional chemistry transport model 'Weather Research and Forecasting coupled with Chemistry' (WRF-Chem) (Grell et al., 2005) in its core. The model's version 3.9.1 has been used. The model domain is centered in Delhi with a horizontal grid spacing of 10 km x 10 km with 50 vertical levels with eight levels in the first 1 km from the surface, and the model top is set at 50 hPa. The simulation uses a time step of 1 minute for temporal integration with radiation calculations done every 12 minutes. The model domain mainly covers the north Indian region spanning from $62\,^0E$ - $93\,^0E$ and $21\,^0N$-$36\,^0N$ (see supplementary figure 1). We use the Rapid Radiative Transfer Model for Global Circulation Models (RRTMG) scheme (Mlawer et al., 1997; Iacono et al., 2000, 2008; Clough et al., 2005) to parameterize the short-wave and long-wave radiative interactions. The choice of the scheme for the parameterization for boundary layer turbulence is vital for the simulations of atmospheric particulate pollutants (Govardhan et al., 2015, 2016, 2019; Sengupta et al., 2022; and the reference therein). The boundary layer processes in the DSS modeling framework are parameterized using the Mellor-Yamada-Nakanishi-Niino 2.5 (MYNN2.5) scheme (Nakanishi and Niino, 2005), which is a turbulent kinetic energy-based scheme that puts a local closure of level 1.5 on the turbulent fluxes. For the parameterization of the microphysical processes, we use the WRF single-moment six-class microphysics scheme (Hong and Lim, 2006). The scheme includes six prognostic water substances, including cloud water, rain, snow, graupel, water vapor, and cloud ice. We parameterize the sub-grid scale convective processes using the Grell-Freitas scheme (Grell and Freitas, 2014). A recent study (Debnath et al., 2022)

highlights the ability of the Grell-Freitas scheme in capturing rainfall characteristics over the Indian region. The DSS uses Noah Land Surface Model (Ek et al., 2003; Niu et al., 2011) to parameterize land-surface processes with the Monin-Obukhov scheme to take into account the surface layer physics (Jiménez et al., 2012). The DSS utilizes the IITM Global Forecasting System model (GFS) to generate the meteorological initial and the boundary conditions for the study domain every 3 hours. This is a global atmospheric model of IITM, Pune, based on the Global Forecasting System of the National Centers for Environmental Prediction (NCEP), USA. The IITM GFS runs in an operational forecasting framework at a horizontal grid spacing of 12 km employing ensemble Karman filtering for assimilating observational data (Mukhopadhyay et al., 2019). The IITM GFS provides the required conditions of the atmospheric state variables like pressure, temperature, winds, specific humidity, etc., to the model domain. The stationary geographic fields like topographical height, surface albedo, land-use, leaf area index, etc., are interpolated from the Moderate Resolution Imaging Spectroradiometer (MODIS) dataset to the model's grid.

**2.2 Anthropogenic Emissions**

We use version 2.2 of the Emission Database for Global Atmospheric Research Hemispheric Transport of Air Pollutants (EDGAR-HTAP) (Janssens-Maenhout et al., 2015) for the prescription of anthropogenic emissions of aerosols and trace gases in the DSS. This global emissions inventory has been constructed by combining multiple regional emission inventories like the Environmental Protection Agency (EPA) for the USA, the European Monitoring and Evaluation Programme (EMEP), and the Netherlands Organisation for Applied Scientific Research (TNO) for Europe, EPA and Environment Canada for Canada, and the Model Intercomparison Study for Asia (MICS-Asia III) for China, India, and other Asian countries. The inventory also provides sector-wise emissions for the five main sectors, including transport, industries, power, residential, and agricultural. The emissions are provided at a spatial resolution of $0.1^0$ in latitude and longitude space. The emissions are available for the aerosols and their precursor gases, including sulfur-di-oxide ($SO_2$), nitrogen oxides (NOx), carbon monoxide (CO), non-methane volatile organic compounds (NMVOC), ammonia ($NH_3$), BC, OC, $PM_{2.5}$, and $PM_{10}$.

For Delhi and the surrounding 19 districts of the National Capital Region (NCR), including Jhajjar, Rohtak, Sonipat, Panipat, Bagpat, Muzzaffarnagar, Meerut, Gautam Buddh Nagar, Faridabad, Ghaziabad, Alwar, Bharatpur, Bulandshahar, Gurgaon, Rewari, Mahendragarh, Rewari, Jind, and Karnal we use the anthropogenic emissions inventory prepared by The Energy and Resources Institute (TERI) for the year 2016. This fine-gridded (4km x 4km) emissions inventory (TERI and ARAI, 2018) provides anthropogenic emissions of $SO_2$, NOx, NMVOC, CO, $PM_{10}$, and $PM_{2.5}$. The $PM_{2.5}$ has been further speciated in OC, BC, Sulphates, Ammonium, Chlorides, and Nitrates. The inventory also provides emissions on a sectoral basis. The sectors could be broadly classified into eight major sectors, including transport, residential, industries, waste burning, construction, road dust, energy, and others (which include the emissions from the sectors like Crematoria, Airports, Restaurants, Non-energy solvent use, and Diesel Generator sets). Moreover, the inventory also includes a monthly variation in emissions from all the aforementioned sectors. For this study, we have re-gridded this emission inventory to a horizontal grid spacing of $0.1^0$ x $0.1^0$ and have subsequently replaced the EDGAR emission fields with this inventory over the NCR region. In general, for Delhi-NCR, there is an increasing trend in the anthropogenic emissions in the recent years. Sahu et al., 2023 reports changes in sectoral emissions over Delhi in 2020 in comparison with 2010. The study suggests that for $PM_{2.5}$, the emissions from transport sector and industries have increased by 37% and 25% respectively. On the other hand, the residential sector emissions show a slight decrease (1-2%). However, due to lack of any such data for the period 2016 to 2022, we have stick to the original inventory of the year 2016 for this study.

For the emissions from agricultural burning activities, we use a combination of the Fire Inventory from NCAR (FINN) database (Wiedinmyer et al., 2011) and the active fire count data from the Visible Infrared Imaging Radiometer Suite (VIIRS) instrument (Schroeder et al., 2014) on-board the Suomi

National Polar-Orbiting Partnership (Suomi NPP) satellite. We have prepared a daily climatology for year-long fire emissions using the FINN data-set for the years 2002 to 2018. On each day of the forecast, we superimpose the near-real time daily active fire count data from the VIIRS instrument on-board the Suomi NPP satellite on the climatological fire emissions file for that day. For day 1 of the forecast, the fire emissions only over those grids are activated where we get non-zero active fire counts on that day

with a confidence level greater than 70%. The other points in the domain are supplied with no fire emissions. For day 2 – day 5 of the forecast, the climatological fire emissions over only those grids are activated where we get non-zero values in the climatological VIIRS fire count data for that day. This dataset is prepared using the VIIRS data the years 2011–2018. Thus, while the Day 1 fire emission forecasts are generated by amalgamation of near-real time fire count and climatological fire emissions,

the Day 2-- Day 5 fire emission forecasts are generated using the climatological information about the fire emissions and the active fire counts. In Supplementary Figure 2, we compare the prescribed emissions of OC from fires in the DSS framework with the corresponding emissions from the long-term climatological data from FINN, and from the Copernicus Atmosphere Monitoring Service (CAMS) Global Fire Assimilation System (GFAS) (Kaiser et al., 2012) for the period of October 2021—November

2021. It may be noted that the fire emissions employed in the DSS framework do show day-to-day variability. They are not overly driven by long-term FINN climatology. However, the peak in the absolute magnitudes of the emissions in DSS looks to reach a week earlier compared to that in GFAS. The fire emissions in DSS and GFAS show a good agreement with the availability of VIIRS fire count information. It is particularly evident around 19th October and 28th October when VIIRS fire counts are

zero and the corresponding prescribed fire emissions in the DSS and GFAS are also zero.

**2.3 Chemical boundary conditions and the mechanism employed**

The boundary conditions for the chemistry variables in DSS are set using the climatological data

from the global chemistry transport model 'Model for Ozone and Related Tracers version 4' (MOZART-4; Emmons et al., 2010). The climatologies are specifically used as the real-time forecast from MOZART-4 is not available. In the future, we plan to replace these climatological boundary conditions using global atmospheric composition forecasts such as the Copernicus Atmosphere Monitoring Service (CAMS) and the Whole Atmosphere Community Climate Model (WACCM). Dynamic chemical lateral boundary

conditions are essential for capturing air pollution events related to dust storms originating outside our domain. The gas-phase chemistry in DSS is simulated using the MOZART-4 chemical mechanism. This mechanism takes into account 85 gas-phase species with 39 photolysis and 157 gas-phase reactions (Emmons et al., 2010). The aerosol processes are simulated by employing the Goddard Chemistry Aerosol Radiation and Transport (GOCART) model that includes five major tropospheric aerosol species,

viz., sulfate, organic carbon (OC), black carbon (BC), dust, and sea salt (Chin et al., 2000, 2002; Ginoux et al., 2001). While sulfate, BC, and OC are simulated as bulk aerosol species, dust and sea salts are resolved into five and four size bins, respectively. The carbonaceous aerosols (BC and OC) are assumed to be present in both the hydrophobic and hydrophilic modes. The conversion of hydrophobic to hydrophilic is assumed to take place with an e-folding lifetime of 2.5 days. The aerosols are assumed to

be deposited down by dry deposition (for all aerosols) and wet deposition (for hydrophilic aerosols) pathways. While it is noted that the GOCART mechanism does not take into account the secondary organic aerosols and the nitrate aerosols, we stick to it as it is computationally less expensive and thus useful in an operational air quality forecasting set-up.

**2.3 Chemical Data Assimilation**

The DSS improves the initialization of aerosol species and thus $PM_{2.5}$ field via assimilation of satellite observations of aerosol optical depth (AOD) using the three-dimensional variational (3DVAR) scheme of the community Gridpoint Statistical Interpolation system (version 3.5). The system assimilates

the observations into the model by minimizing the cost function $J(x)$ (equation 1), which is the sum of the deviation of the final state of the model from its background state and the observations. The cost function takes the following form,

$$J(x) = \frac{1}{2}(x - x_b)^T B^{-1} \quad (x - x_b) \quad + \quad \frac{1}{2}(H(x) - y)^T R^{-1} \quad (H(x) - y) \quad \text{.... (1)}$$

Where $x$ is the state vector which is composed of aerosol chemical composition and meteorological parameters needed for AOD calculation, $x_b$ is the information about $x$ available prior to the assimilation (also known as background information), $B$ is the background error covariance (BEC) matrix, $H$ is the forward operator that calculates AOD from the WRF-Chem aerosol chemical composition following Liu et al. (2011), $y$ is the AOD retrieved by MODIS, and $R$ is the observational error covariance matrix. More details about each of the terms in equation 1 can be found in Kumar et al. (2020). The assimilation of MODIS AOD (from both TERRA and AQUA satellites) in the model is done at 9 UTC every day in the DSS. In addition to assimilation of satellite data, we also assimilate surface measurements of $PM_{2.5}$ into the model at 9 UTC. The data comes from 43 stations of the Central Pollution Control Board (CPCB) and the Delhi Pollution Control Committee (DPCC), spanned across Delhi. The exact names and the locations of the stations can be found in supplementary figure 1 of Sengupta et al. (2022).

**2.4 Tagged-tracers in DSS**

We have added a variety of passive tagged-tracers in WRF-Chem, which assist us in understanding the region- and source-specific contribution to $PM_{2.5}$ mass concentration over Delhi. The passive tracer of a regular species is that species introduced in the model which undergoes all physio-chemical processes identical to a regular chemical species (e.g., emissions, transport, chemical transformation, deposition, etc.) without providing feedback to the model (Bhardwaj et al., 2021; Kumar et al., 2015). In other words, the tracer species does not take part in radiation or droplet formation processes, as its effect in such feedback processes is already taken into account by the parent regular species. The difference between a regular chemical species and a tracer chemical species is illustrated in fig.1.

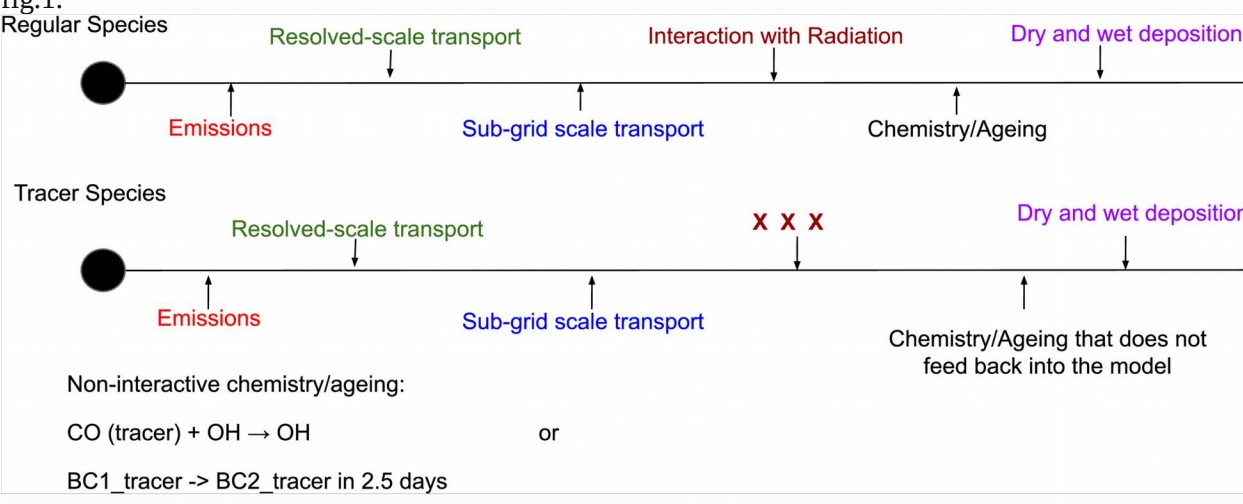

Figure 1: The life-cycle of a regular chemical species and a tracer chemical species is illustrated. The main difference lies in the feedback and the chemistry sections. The tracer species does not have feedback

on the radiation processes in the model, and it does not affect the chemistry of regular species in the model. Two such examples of non-interactive chemistry are given. The CO tracer species gets oxidized by OH⁻ radical, but it does not change the mass budget of the OH⁻ radical in the model. Similarly, the tracer hydrophobic BC (BC1_tracer) species gets aged into the tracer hydrophilic BC species (BC2_tracer) while keeping the mass of the regular hydrophilic BC in the model intact.

Since $PM_{2.5}$ is not a prognostic species in the model, we employ tracers for hydrophobic black carbon (BC1), hydrophilic black carbon (BC2), hydrophobic organic carbon (OC1), hydrophilic organic carbon (OC2), non-speciated primary $PM_{2.5}$ (P25), and carbon monoxide (CO). The GOCART scheme employed in the WRF-Chem model used in this study calculates $PM_{2.5}$ as follows,

$$PM_{2.5} = BC1 + BC2 + (OC1 + OC2) \times 1.8 + P25 + DUST1 + SEAS1 +$$
$$(0.286 \times DUST2) + (0.942 \times SEAS2) + 1.375 \times Sulfate \quad \quad .... (2)$$

Where,

$DUST1$ = Mineral dust aerosol species falling in the first bin with the effective radii equal to 0.73 μm
$DUST2$ = Mineral dust aerosol species falling in the second bin with the effective radii equal to 1.4 μm
SEAS1= Sea-salt aerosol species falling in the first bin with the effective radii equal to 0.3 μm
SEAS2= Sea-salt aerosol species falling in the second bin with the effective radii equal to 1.0 μm
Sulfate= Sulfate aerosol species,
In this study, we employ tracers for five of the ten species involved in the calculation of $PM_{2.5}$ in the GOCART scheme (equation 2). In figure 2, we examine the contribution of those ten species to the simulated $PM_{2.5}$ in the model.

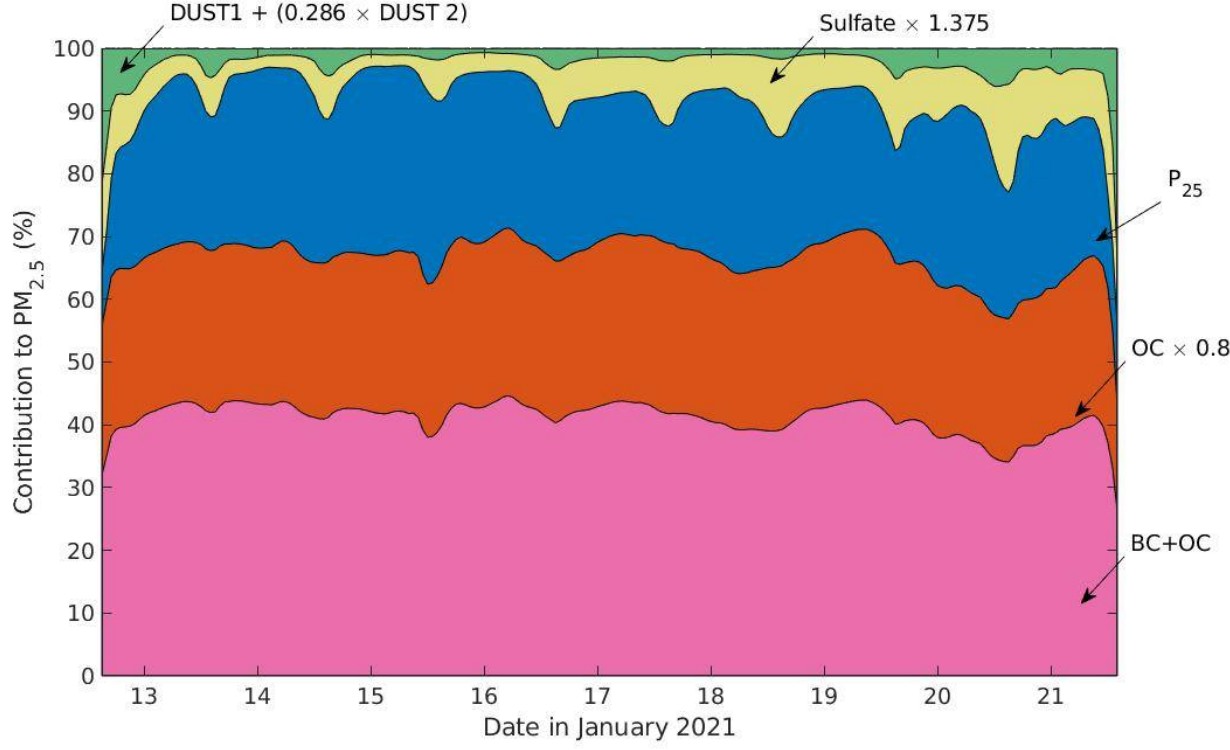

Figure 2: Speciation of the WRF-Chem simulated near-surface $PM_{2.5}$ mass concentration over Delhi during January 2021. Contribution from SEAS1 and SEAS2 to $PM_{2.5}$ in Delhi is negligible during the

study period and thus it is not shown in the figure.

It may be noted that the chosen five species (BC1, BC2, OC1, OC2, and P25) together contribute around 85-90% of the total $PM_{2.5}$ in the model. Thus, our five tracers would together represent, on an average, 85-90% of the corresponding $PM_{2.5}$ mass concentrations. Therefore, practically we can interpret those five tracers together as a $PM_{2.5}$ tracer. Adding tracers for $SO_4^{--}$, DUST1, DUST2, SEAS1, and SEAS2 would not drastically affect the overall results as their contribution to $PM_{2.5}$ over Delhi, specifically during the winter season, is negligible, especially in the model simulations (however, the fractional contribution of different species during April-September could be different due to dust storms and monsoon circulation affecting this region). Moreover, since the forecasting system is operational on a daily basis, one needs to limit the computational load and thus the total number of species in the model configuration to keep avoid daily run-time as short as possible. Keeping all these constraints in mind, we chose to put tracers only for the five selected species.

### 2.4.1 Tracers for Anthropogenic PM$_{2.5}$ in the model
We introduce regional tracers for the total emitted anthropogenic $PM_{2.5}$ from Delhi and the 19 districts surrounding it. These districts, along with Delhi, form the NCR. The following are the districts included: Delhi, Jhajjar, Sonipat, Bagpat, Ghaziabad, Gautam Buddha Nagar, Faridabad, Gurgaon, Rohtak, Jind, Panipat, Karnal, Muzaffarnagar, Meerut, Bulandshahr, Bharatpur, Alwar, Mahendragarh, Rewari, and Bhiwani. In figure 3, we show the locations of these 20 districts.

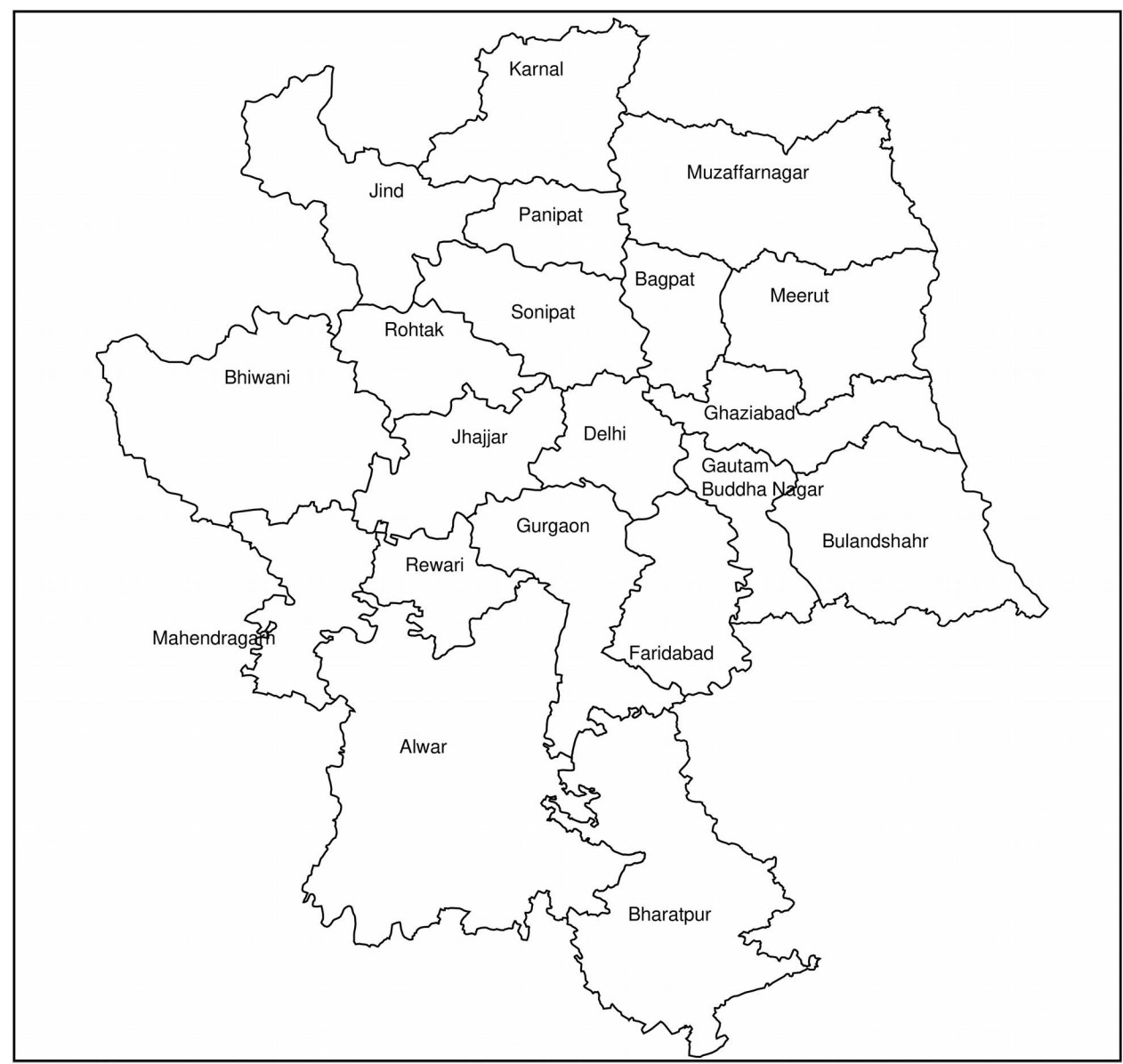

Figure 3: The locations of the 20 districts of NCR whose anthropogenic $PM_{2.5}$ emissions are tagged in
DSS.

In addition to  those 20 districts, we also trace $PM_{2.5}$ from eight broad source-based categories
exclusively in Delhi. These individual broad categories are a group of several sub-categories put together.
The broad categories and the included sub-categories are listed in table 1. As mentioned in section 2.2, the
emissions inventory provides extensive sub-categorical information for the entire NCR domain. However,
version 1.0 DSS does not trace the $PM_{2.5}$ emissions from the individual broad categories from the NCR
districts other than Delhi. Even for Delhi, the emissions from the individual sub-categories are not traced.
All these ensure the computational speed and cost for the operational DSS system. Moreover, the tagged
sources fulfill the current requirements of the policymakers with regards to the air quality managment in
the city.

| Broad categories | Included sub-categories |
|---|---|
| Transport | Diesel vehicles, Gasoline vehicles, and CNG vehicles |
| Industries | Industries, stone crushers, Brick industry, and Refineries |
| Construction | Construction activities |
| Road dust | Dust emissions from paved roads |
| Waste burning | Refuse burning, Landfill fires, and Incinerators |
| Energy | Power Plants in NCR, Badarpur power plant in Delhi, and Flyash ponds |
| Residential | Domestic-biomass, and other fuels |
| Others | Crematoria, Airport, Restaurant, Non-energy solvent use, and Diesel Generator sets |

Table 1: The source-based $PM_{2.5}$ tracers employed only for Delhi in this version of DSS. It is to be further noted that we employ tracers for the eight broad source categories (column 1) in Delhi. We do not employ tracers for the individual sub-categories in this version of DSS.

### 2.4.2 Tracers for biomass-burning activities

Along with the anthropogenic emissions of $PM_{2.5,}$ we also trace the biomass-burning generated emissions of $PM_{2.5}$. Similar to the anthropogenic $PM_{2.5}$, we introduce tracers for biomass-burning generated BC1, BC2, OC1, OC2, and P25. These tracers hold significant importance in DSS, as the post-monsoonal harvesting of paddy generates a large amount of stubble which gets burnt and generates a thick layer of smoke in the upwind regions of Delhi, which eventually travels to Delhi. So, the tracers representing those burning activities help us identify the contribution of biomass-burning to the $PM_{2.5}$ load in Delhi and thus are critical for air quality management in Delhi.

### 2.4.3 Scenario tracers for Anthropogenic $PM_{2.5}$

Apart from tracing the anthropogenic and the biomass-burning generated $PM_{2.5}$, DSS offers a very unique feature, which we term 'scenario tracers'. The scenario tracers are very similar to the other anthropogenic $PM_{2.5}$ tracers, with the main difference laying in the emission magnitudes of these tracers. In DSS, a scenario tracer of a regular species has its emission 20 or 40% lesser than the regular species. Therefore, the scenario tracer represents a scenario in which the emissions of the corresponding regular species are reduced by 20 or 40%. We have introduced these scenario tracers for all the 20 districts and all the eight broad source categories in Delhi. These scenario tracers play a vital role in guiding the authorities about the possible effects of the source-level interventions. The advantage of scenario tracers is that it gives an opportunity to generate numerous emission reduction scenarios, which would guide the policy-makers in finalizing the intervention targets. The use of these tracers for air quality management purposes will be shown in the results section.

### 2.4.4 Chemical data-assimilation for tracers

Another important feature of DSS is chemical data-assimilation applied for the tracer species. In DSS, for every grid point in the model domain, we identify the ratio by which the regular species like BC1, BC2, OC1, OC2, and P25 are modified due to the assimilation of satellite as well as ground-based data. We multiply all the corresponding tracers species by the same ratios to get them closer to reality.

## 2.5 Post-processing of the output

With the aforementioned tracers of different categories, we introduce a total of 470 new tracers in WRF-Chem for the purpose of DSS. Upon running DSS in an operational forecasting setup, we generate
an enormous amount of data that needs to be processed to get meaningful information. In the post-processing and analysis of the output, we extract the surface level data for all the tracers and the main regular species. Since our focus of analysis is Delhi, we mask out all other regions from the variable fields. By doing this, we estimate the contribution of $PM_{2.5}$ emitted from all the regions of interest to $PM_{2.5}$ in Delhi. Moreover, we also get to know the contribution from the sources in Delhi to $PM_{2.5}$ in
Delhi. The change in $PM_{2.5}$ due to the emission reduction scenarios is subsequently found. All the analysis is made publicly available daily at https://ews.tropmet.res.in/dss/.

## 2.6 Overall flow of DSS

Figure 4 depicts the operational functioning of DSS. The input data needed for the chemistry part
(white boxes, fig.4), i.e., the anthropogenic and biomass-burning emissions and chemical boundary conditions, are generated using the utilities like anthro, FINN, and mozbc as explained in sections 2.2 and 2.3. Note that biomass-burning emissions are generated using FINN and the VIIRS active fire count data. The meteorological input component (white boxes with a circle in their left corner, fig.4) consists of the meteorological boundary forcing data (IITM GFS model output) and the stationary geographical data,
both of which are processed by the WRF Preprocessing System (WPS) to create the model compatible input and boundary forcing. Both the chemistry and meteorological input data are then processed by the core part of the DSS (gray boxes, fig.4) to create the initial and the boundary condition files. Subsequently, DSS carries out the chemical data assimilation using the CPCB and the satellite data (gray blocks with a circle in their right corner, fig.4). After this step, the actual WRF-Chem run with 400 tracers
is carried out for the next five days. Upon the completion of the simulation, the outputs are suitably post-processed to generate two main results (gray boxes with a rectangle in their right corner, fig.4) a) source apportionment of $PM_{2.5}$ in Delhi to understand the contribution of the surrounding 19 districts and the eight sectors in Delhi, and b) the effects of the various emission reduction scenarios on $PM_{2.5}$ in Delhi. The results are then sent to the governing and decision-making authorities, which could take certain
policy-level decisions in order to manage the air quality in Delhi. If the decision-making authorities decide to carry out certain source-level interventions (e.g., Ghude et al., 2022), then those interventions are then incorporated into the DSS through the feedback section (gray block with a triangle in its right corner, fig.4).

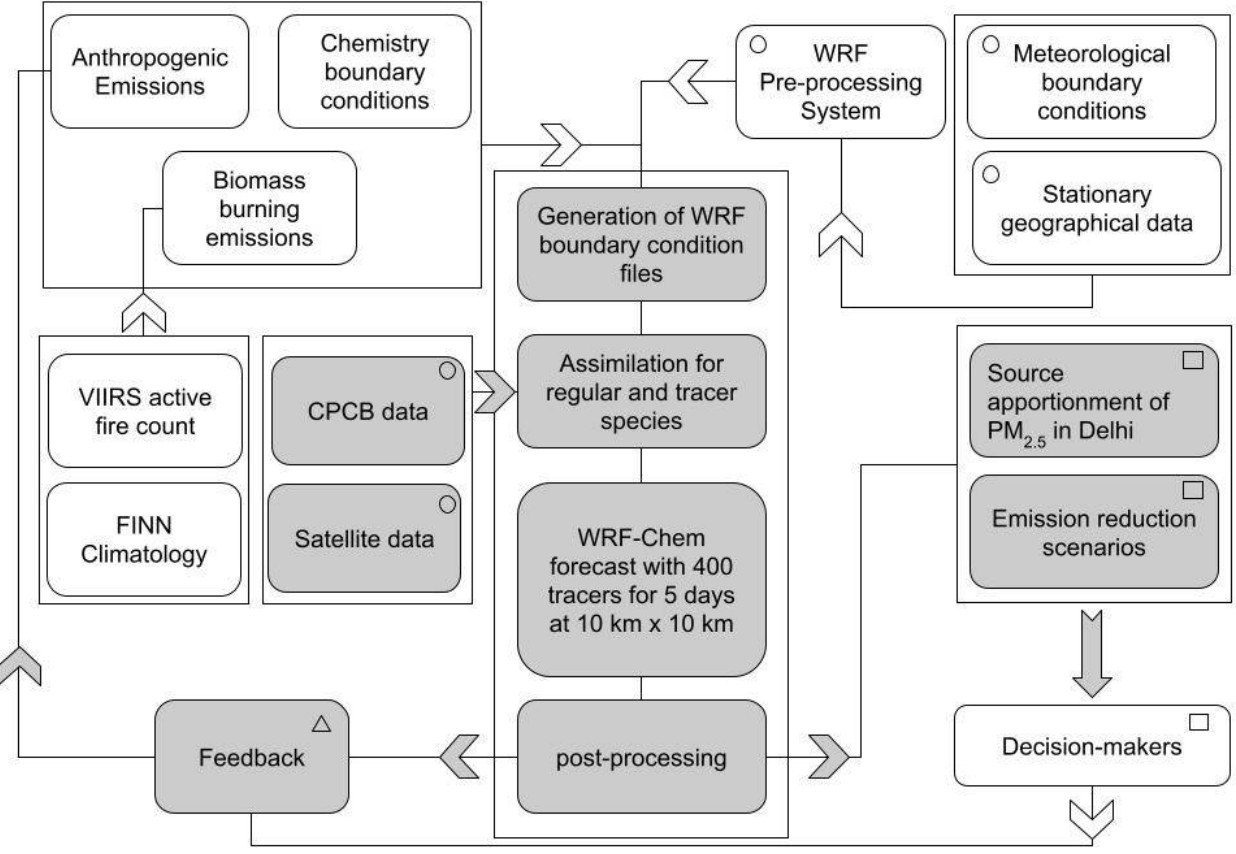

Figure 4: Block diagram for DSS: The white boxes denote input data needed for the chemistry part, the white boxes with a circle in their left corner stand for the input data related to the meteorological component. The gray blocks represent the core part of DSS, which is mainly related to the running of the WRF-Chem model. The gray blocks with a circle in their right corner denote the input data needed for chemical data-assimilation purposes. The gray boxes with a rectangle in their right corner stand for the standard outputs from the DSS, which are communicated to the decision-makers (white block with a rectangle in its right corner). The feedback (gray block with a triangle in its right corner) from the decision-makers and the model's post-processed output are analyzed, and accordingly, the emissions of the anthropogenic activities are modified. A more detailed explanation of the working principle of each block can be found in sections 2 and section 2.6.

**3. Results and discussion:**

**3.1 Performance evaluation for DSS:**

We examine the DSS-simulated near-surface $PM_{2.5}$ mass concentration against the corresponding observations carried out at the CPCB and DPCC stations in Delhi. We divide the entire period of 5 months into the post-monsoon (October–November) and winter (December–February) seasons as the stubble-burning activities are prevalent mainly during the post-monsoonal season, while the winter season pollution is primarily governed by the local as well as distant anthropogenic emissions and the pollution-conducive meteorology. Thus, such a division is essential to  help us understand the performance of DSS

in capturing the season-specific emission sources and the associated pollutants' concentrations. We evaluate the performance of DSS for Day 1 to Day 5 of every day's forecasts. During the post-monsoonal period (fig. 5a), the simulated daily-mean $PM_{2.5}$ closely matches the measurements for the month of October 2021. The sharp reduction in the $PM_{2.5}$ during mid-October (17–20 October) is well captured by the model for all the lead times (i.e., Day 1 to Day 5). In the first week of November (black cricles, fig.5a

and fig.5c), the model shows a large underestimation with respect to the observations. This period was mainly associated with the peak of stubble-burning activities (Govardhan et al., 2022) and the Diwali festival in 2021. Both these events result in emissions of a significant amount of particulate pollutants and their precursor gases (Singh et al., 2010; Parkhi et al., 2016; Cusworth et al., 2018; Chowdhury et al., 2019; Kulkarni et al., 2020; Saxena et al., 2020). The large uncertainty associated with both these

emission sources (Vadrevu et al., 2015; Liu et al., 2018; Mukharjee et al., 2020; Kumar et al., 2020) results in the under-estimated $PM_{2.5}$ mass concentrations by DSS. The improvements in emission inventories such as the use of Fire Radiative Power (FRP) for estimating and temporally allocating fire emissions, incorporation of emissions from fire crackers would help improve the estimates. On the contrary, the model simulations over-estimate $PM_{2.5}$ during the following week. Owing to the persistent

severe air pollution days and a forecast of a similar scenario from $15^h$–$19^{th}$ November 2021, the Government of Delhi and the CAQM had issued certain restrictions on the traffic in the city, banned construction activities, ordered remote schooling and working guidelines, and had banned the entry of the heavy vehicles into the city (CAQM 2021). As a result, the $PM_{2.5}$ concentration in the city showed a reduction in the following week. The simulations did not implement such restrictions in the modeling

framework and thus overestimated the $PM_{2.5}$ concentration during this week. We further note that the fire activity in the neighboring states of Punjab and Haryana during that period was also on a declining trend (Fig.1, Govardhan et al., 2023), so the associated fire emissions may not be completely responsible for this behavior of the model. For the entire duration, the mean overstimation is found to be 21.94%. This overestimation is consistent with the previous estimation put forth by Ghude et al., 2022. Towards the end

of November, the model captures the day-to-day variations in the observed $PM_{2.5}$ but underestimates the actual magnitudes. Such a behavior could be associated with the coarse grid-spacing of the model (10 km), which limits its ability to simulate higher PM concentrations. For the $AQI_{PM2.5}$ (fig.5b), the model has more tendency to generate AQI up to 300 (barring the episode of $15^{th}$–$19^{th}$ November 2021). The disagreements with observations in $PM_{2.5}$ get reflected in the AQI as well. It may well be noted that the

model's performance does not drastically degrade from Day 1 to Day 5. A detailed analysis of the model's ability to capture $PM_{2.5}$ and the associated AQI has been shown in tables 2–5, which will be discussed further.

    For the winter period (fig.5c), DSS shows a better agreement with the observation up to the mid of December, beyond which the model starts to under-perform in comparison with the observations. The

model simulations are capable of simulating the $PM_{2.5}$ concentrations as high as 200 µg m$^{-3}$ however, they are not able to simulate the values greater than that. Improvements in the emission inventory would be vital to achieve that. This issue is likely to be related to the coarser grid spacing in the simulations, unrealistic simulations of meteorological parameters (like the planetary boundary layer height, near-surface winds, etc.) (Govardhan et al., 2015, 2016), and limitations associated with the chemistry scheme

in the model which may not adequately represent the ambient air pollution chemistry in Delhi (Jena et al., 2020; Pawar et al., 2022), and under-representation of the emission sources in the region due to the unavailability of the real-time dynamic emissions inventory (Sengupta et al., 2022). The current emissions inventory used in the model though does have some information about the sources like open waste burning and brick kilns, in and around Delhi, this information is likely to be underestimating the

reality in 2021. The emissions inventory employed in this study was compiled using surveys done in 2016. There are significant changes that have occurred in the emissions magnitudes from 2016 to 2021. We note that these uncertainties will affect the model simulations. Moreover, during the January month the temperatures of the region fall down. The residents of Delhi burn biomass or solid wood for space

heating purposes. Such sources are missing in the employed emissions inventory. Additionally, such burning activities occur at a very fine spatial scales which can not be identified by remote sensing techniques. Thus, a part of the underestimation during the month of January would be related to these factors. In addition to this, the lower temperatures bring foggy conditions into the picture. Such weather conditions promote a large number of atmospheric chemical reactions resulting in gas-to-particle conversion of volatile gas phase species into secondary aerosols. Such processes are currently missing the models' chemical mechanism. This further enhances the underestimations in the model. All these factors put together result in the underestimated $PM_{2.5}$ in the model vis-a-vis the measurements. Nonetheless, DSS does a better job in the month of February when the ambient $PM_{2.5}$ concentration is mostly below 200 $\mu g\ m^{-3}$. The $AQI_{PM2.5}$ is also better captured in the winter season (fig. 5d) compared to the post-monsoon period (fig.5b). The model does capture some events of very poor AQI conditions ($300 < AQI \leq 400$). However, the severe AQI values (AQI > 400) are missed by the model. Overall, the model captures the air quality conditions up to the very-poor AQI category, but it can not quantitatively capture the severe air pollution events. However, it is also to be noted that during an observed severe air pollution event (AQI > 400), the simulated AQI lies only one category below (i.e., in the very poor AQI category). Thus, the model does show signatures of severe air pollution but fails to capture the actual magnitudes. It may also be noted that whenever the modeled AQI is in or above the very-poor category, the observed AQI almost always lies in or above the very-poor category, i.e., our system is able to capture extreme events very well.  This point is illustrated further in tables 4 and 5. The supplementary figure 3 clearly depicts that the simulated AQI captures the overall trend of the observed AQI, however the magnitudes of AQI are not captured by the model.

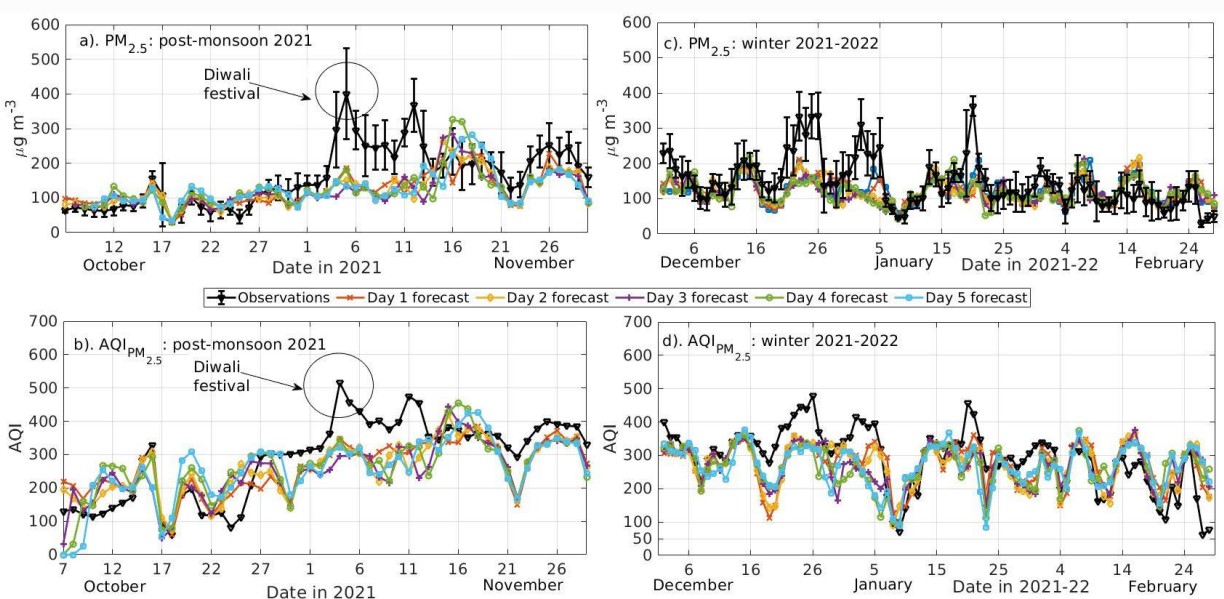

Figure 5: Performance of the DSS in simulating near-surface $PM_{2.5}$ mass concentration ($\mu g\ m^{-3}$) over Delhi in comparison with the observations averaged over the 39 observational locations across the city. a). Model Vs Observation comparison for the simulated daily mean $PM_{2.5}$ mass concentration during the post-monsoonal season of 2021. The error bars on the black line indicate the one standard deviation range for the observations. b).  Model Vs Observation comparison for the daily mean AQI associated with $PM_{2.5}$ during the post-monsoonal period c). similar comparison as a, for the winter season. d). similar to b, for the winter period. The black circles mark the days of Diwali festival during the post-monsoon period of 2021.


We further compute the relevant statistical parameters, namely mean bias (MB), mean error (ME), root mean square error (RMSE), normalized mean bias (NMB), normalized mean error (NME), fractional bias
(FB), and fractional error (FE) for the model-observation comparison of the near-surface $PM_{2.5}$ mass concentration for post-monsoon 2021 (table 2) and winter 2021-22 (table 3). We report the statistics individually for moderate ($100 < AQI \leq 200$), poor ($200 < AQI \leq 300$), and 'very poor and above' ($AQI > 300$) AQI categories for Day 1 to Day 5 forecasts. The formulae used for calculating the statistical parameters are listed in section 4 of the supplementary material. For the post-monsoon season (table 2),
DSS shows the least MB under poor AQI conditions. Expectedly, ME and RMSE are higher for very poor and above AQI categories. Moreover, they gradually increase from Day 1 to Day 5 forecasts for all the scenarios. Nevertheless, the change in ME or RMSE from Day 1 to Day 5 is within 30% of the ME or RMSE of Day 1 forecasts, especially for the very poor and above AQI conditions. This signifies the accuracy of the forecasts over a longer time horizon. The NMB and NME values are limited to ±0.30 and
±0.50, suggesting that DSS depicts an acceptable accuracy for the simulated $PM_{2.5}$ mass concentrations (Emery et al., 2017)  for all the AQI categories through Day 1 to Day 5 forecasts. Specifically, NMB (NME)  values do not cross 0.1 (0.37) for the poor AQI category,  highlighting the accuracy of DSS and its ability to match the best model in the community (Emery et al., 2017). Like MB and NMB, FB is the least for the poor AQI conditions. The DSS tends to over-predict (under-predict) the $PM_{2.5}$ with positive
(negative) MB, NMB, and FB values during moderate (poor and above conditions) AQI conditions. Nevertheless, the system can simulate the observed $PM_{2.5}$ during the post-monsoonal months with an acceptable deviation (Emery et al., 2017), especially when the observed AQI is in the poor or above categories.





| AQI category | Day | MB (μg m⁻³) | ME (μg m⁻³) | RMSE (μg m⁻³) | NMB | NME | FB | FE |
|---|---|---|---|---|---|---|---|---|
| Moderate | Day 1 | 16.34 | 18.48 | 25.18 | 0.23 | 0.26 | 0.20 | 0.23 |
| | Day 2 | 13.37 | 17.24 | 23.87 | 0.19 | 0.24 | 0.17 | 0.22 |
| | Day 3 | 17.61 | 20.93 | 27.70 | 0.23 | 0.28 | 0.22 | 0.26 |
| | Day 4 | 24.52 | 27.51 | 36.60 | 0.30 | 0.33 | 0.29 | 0.33 |
| | Day 5 | 24.02 | 26.84 | 33.73 | 0.27 | 0.30 | 0.28 | 0.32 |
| | | | | | | | | |
| Poor | Day 1 | -19.97 | 31.99 | 38.61 | -0.17 | 0.28 | -0.19 | 0.31 |
| | Day 2 | -17.27 | 27.85 | 37.18 | -0.15 | 0.24 | -0.16 | 0.26 |
| | Day 3 | -20.49 | 27.18 | 36.51 | -0.18 | 0.24 | -0.20 | 0.26 |
| | Day 4 | -10.53 | 27.25 | 37.00 | -0.09 | 0.24 | -0.10 | 0.25 |
| | Day 5 | -1.20 | 29.20 | 40.08 | -0.01 | 0.26 | -0.01 | 0.26 |
| | | | | | | | | |
| Very poor and above | Day 1 | -65.59 | 76.10 | 101.26 | -0.31 | 0.36 | -0.37 | 0.42 |
| | Day 2 | -67.11 | 79.76 | 110.05 | -0.32 | 0.38 | -0.38 | 0.45 |
| | Day 3 | -72.82 | 86.71 | 115.57 | -0.34 | 0.41 | -0.42 | 0.49 |
| | Day 4 | -67.02 | 85.77 | 107.39 | -0.32 | 0.40 | -0.38 | 0.48 |
| | Day 5 | -68.11 | 85.06 | 112.84 | -0.32 | 0.40 | -0.38 | 0.48 |

Table 2: The statistical parameters associated with the model evaluation for the simulated near-surface PM$_{2.5}$ mass concentration for the post-monsoonal season of 2021. The meaning of the acronyms can be found in section 3.1. The ideal value for all the statistical parameters is zero. The units of MB, ME, and RMSE are μg m⁻³, while the other parameters are unitless.

For the winter season of 2021-22, the MB values for the moderate category (table 3) are twice that of the post-monsoonal period, indicating a higher overestimation of the moderate AQI conditions in the model in the winter period. On the other hand, the MBs for poor and 'very poor and above' AQI scenarios are comparable to that in the post-monsoonal months. The ME, RMSE, and NME remain roughly the same for Day 1 through Day 5 forecasts, which increases the trustworthiness of the forecasts on short to medium-range time scales. Similar to the post-monsoon season, the NMB and NME values for the winter season are lesser than ±0.3 and 0.5, respectively, underscoring the ability of the system to capture the observed PM$_{2.5}$ mass concentrations very adequately (Emery et al., 2017). Similarly, for the 'poor' AQI category, the NMB and NME values are less than ±0.1 and 0.35, respectively, suggesting an outstanding performance by DSS in this category (Emery et al., 2017). It is to be noted that the NMB values for the 'very poor and above' scenarios are higher compared to the poor scenario. This is likely because the 'very poor and above' category holds a broader range of AQI values (AQI > 300) compared to the 'poor' AQI bracket (200 < AQI ≤ 300), which results in the higher NMB in the former compared to the latter. Similar to the post-monsoonal period, the system has a tendency to overestimate (underestimate) the PM$_{2.5}$ under moderate (very poor and above) AQI conditions, which is reflected in the positive (negative) MB, NMB, and FB values. Overall, the performance of the DSS is improved in the winter season compared to the post-monsoonal season (indicated by the lower values of the relevant statistical parameters in table 2 and table 3).

| AQI category | Day | MB ($\mu g\ m^{-3}$) | ME ($\mu g\ m^{-3}$) | RMSE ($\mu g\ m^{-3}$) | NMB | NME | FB | FE |
|---|---|---|---|---|---|---|---|---|
| Moderate | Day 1 | 22.25 | 32.66 | 44.25 | 0.29 | 0.43 | 0.26 | 0.37 |
| | Day 2 | 20.43 | 32.56 | 42.38 | 0.27 | 0.43 | 0.24 | 0.38 |
| | Day 3 | 25.78 | 36.70 | 49.92 | 0.34 | 0.48 | 0.29 | 0.41 |
| | Day 4 | 24.29 | 35.50 | 47.93 | 0.32 | 0.47 | 0.28 | 0.40 |
| | Day 5 | 22.31 | 34.92 | 44.73 | 0.29 | 0.46 | 0.26 | 0.40 |
| | | | | | | | | |
| Poor | Day 1 | 4.50 | 27.20 | 34.50 | 0.04 | 0.26 | 0.04 | 0.26 |
| | Day 2 | 6.75 | 29.95 | 40.25 | 0.06 | 0.29 | 0.06 | 0.28 |
| | Day 3 | 7.40 | 33.96 | 44.84 | 0.07 | 0.33 | 0.07 | 0.32 |
| | Day 4 | 3.75 | 33.58 | 42.16 | 0.04 | 0.32 | 0.04 | 0.32 |
| | Day 5 | 4.84 | 34.70 | 44.99 | 0.05 | 0.33 | 0.05 | 0.33 |
| | | | | | | | | |
| Very poor and above | Day 1 | -58.66 | 75.54 | 97.63 | -0.28 | 0.36 | -0.33 | 0.42 |
| | Day 2 | -60.70 | 76.96 | 101.21 | -0.29 | 0.37 | -0.34 | 0.43 |
| | Day 3 | -65.98 | 83.00 | 107.70 | -0.32 | 0.40 | -0.38 | 0.47 |
| | Day 4 | -67.49 | 80.42 | 106.76 | -0.32 | 0.39 | -0.39 | 0.46 |
| | Day 5 | -65.21 | 80.20 | 106.45 | -0.31 | 0.39 | -0.37 | 0.46 |

Table 3: Similar to table 2 but for the winter period of 2021-22.


We have also examined the ability of DSS to capture the AQI associated with $PM_{2.5}$ mass concentration values in comparison with the corresponding observations. To assess the model's performance, we have computed the statistical parameters, namely Accuracy, False Alarm Ratio (FAR), Probability of Detection (POD), Critical Success Index (CSI), Success Ratio (SR), and Bias. These

parameters are calculated for the individual AQI categories using the contingency table and the formulae presented in section 5 of the supplementary material. From table 4, it can be seen that, during the post-monsoon season, the Accuracy is generally high for all the AQI scenarios. For the poor and moderate categories, this could be an artifact of the correct forecasts of the non-events, while for the 'very poor and above' AQI category, this behavior could be attributed to the correct forecasts for both the events and the

non-events (fig.5b). Please note that here the 'event' (non-event) refers to the occurrence (non-occurrence) of the observed AQI in the desired AQI range. The Probability of Detection (POD) comprehends the ability of the model in giving correct forecast for occurrence of an event. On the other hand, 'Accuracy', describes the ability of the model in giving correct forecast of an event or a non-event too. Thus, Accuracy encompasses the event and non-event space, while POD cover only the event space.

For the 'Poor' AQI category, it may be noted that during the post-monsoonal season (fig.5b) after 27[th] October 2021, the observed AQI is always greater than 200 i.e. above the 'poor' category. Thus, as far as the 'Poor' AQI category is concerned, all those instances are recognized as 'non-events'. The model simulated AQI on most of those instances (if not all) is seen to be greater than 200, thus correctly giving forecasts of non-event. This correct forecasts of non-events mainly results in respectable value of

Accuracy for the model forecasts as far as 'Poor' AQI category is concerned. On the other hand, from 27[th] October 2021 to 30[th] November 2021, the POD for 'Poor' AQI does not exist as the observed AQI does not exist in 'Poor' category. Prior to 27[th] October, the Observed AQI does exist in the 'Poor' category, the model forecasts for Day-4 and Day-5 fail to capture that on certain occasions. This failure results in lesser POD for Day-4 and Day-5 forecast in capturing AQI in 'Poor' Category.

The FAR is higher for moderate and poor categories suggesting false forecasts of the non-events; this could be partly related to the fact that the model-simulated AQI does not reach the very poor and above categories as frequently as the observations but remains in the poor category on more instances as compared to the observations. This results in a higher FAR for the poor category. On the other hand, the FAR for the 'very poor and above' AQI category is drastically low, which enhances the confidence in the

simulated AQI in the very poor and above category. The POD is low for the poor and moderate, while it is relatively higher for the 'very poor and above' category. The CSI values, which indicate the overall success of the forecasting system, are relatively high for the 'very poor and above' category and lower for the poor category. Thus, during the post-monsoon season, DSS shows trustworthy performance for the AQI ranging beyond very-poor conditions.


| AQI category | Day | Accuracy (%) | FAR (%) | POD (%) | CSI (%) |
|---|---|---|---|---|---|
| Moderate | Day 1 | 75.62 | 50.99 | 39.87 | 28.18 |
|  | Day 2 | 82.41 | 35.64 | 59.81 | 44.93 |
|  | Day 3 | 79.86 | 41.20 | 53.70 | 39.02 |
|  | Day 4 | 77.55 | 45.76 | 41.16 | 30.55 |
|  | Day 5 | 74.61 | 55.56 | 23.15 | 17.96 |
| Poor | Day 1 | 67.67 | 86.76 | 59.79 | 12.16 |
|  | Day 2 | 69.75 | 83.91 | 72.16 | 15.15 |
|  | Day 3 | 63.35 | 87.80 | 62.89 | 11.38 |
|  | Day 4 | 65.12 | 92.57 | 31.96 | 6.42 |
|  | Day 5 | 60.88 | 95.35 | 21.65 | 3.98 |
| Very Poor and above | Day 1 | 80.86 | 0.36 | 69.48 | 69.31 |
|  | Day 2 | 78.86 | 0.00 | 66.01 | 66.01 |
|  | Day 3 | 72.07 | 0.00 | 55.09 | 55.09 |
|  | Day 4 | 72.92 | 9.30 | 62.90 | 59.09 |
|  | Day 5 | 70.06 | 13.46 | 61.41 | 56.06 |

Table 4: The statistical parameters associated with the evaluation of the simulated AQI associated with $PM_{2.5}$ mass concentration for the post-monsoonal season of 2021. The meaning of the acronyms can be found in section 3.1, and details about the formulae are mentioned in section 5 of the supplementary material. The ideal values for Accuracy, FAR, POD and CSI are 100.0, 0.0, 100.0, and 100.0, respectively.


For the winter season (table 5), the model's behavior roughly remains the same as the post-monsoon, with the only difference occurring in the poor AQI category. The FAR for the 'poor' category drops with a consequent increase in CSI. Nevertheless, the model still behaves the best when AQI goes to 'very poor' and above, with FAR limiting only to as high as 21% and the POD and CSI crossing 60%.

Thus, the analysis assures that the model-simulated AQI is trustworthy for values beyond 300.

The Graded Response Action Plan (GRAP) includes a variety of predefined temporary emission

control measures for all the PM$_{2.5}$ and PM$_{10}$ AQI categories. Expectedly, the GRAP regulations become more stringent when the AQI goes beyond very poor and above (CAQM, 2022). Starting from October 2022, the GRAP in Delhi will be made operational based on the AQI forecast released by the air quality forecasting models (CAQM, 2022). The low FAR for DSS in the 'very poor and above' categories certainly increases the confidence about the simulated AQI in this range and thus permits us to use the model data to implement GRAP in the city. Additionally, the FAR values for the 'very poor and above' categories remain within 20% for day one to day five forecasts for both seasons. This further assures the use of short to medium-range DSS forecasts for implementation of GRAP when AQI goes beyond very-poor conditions.

| AQI category | Day | Accuracy (%) | FAR (%) | POD (%) | CSI (%) |
|---|---|---|---|---|---|
| Moderate | Day 1 | 82.35 | 65.39 | 53.13 | 26.51 |
| | Day 2 | 83.29 | 62.29 | 60.55 | 30.27 |
| | Day 3 | 84.46 | 62.18 | 46.09 | 26.22 |
| | Day 4 | 84.13 | 68.78 | 26.95 | 16.91 |
| | Day 5 | 84.41 | 65.98 | 32.03 | 19.76 |
| Poor | Day 1 | 69.34 | 63.43 | 46.22 | 25.65 |
| | Day 2 | 70.41 | 60.41 | 55.62 | 30.09 |
| | Day 3 | 60.86 | 70.01 | 53.17 | 23.72 |
| | Day 4 | 57.49 | 72.17 | 53.78 | 22.46 |
| | Day 5 | 59.55 | 70.23 | 56.44 | 24.21 |
| Very Poor and above | Day 1 | 76.03 | 15.47 | 73.44 | 64.74 |
| | Day 2 | 75.66 | 12.52 | 69.30 | 63.04 |
| | Day 3 | 68.49 | 17.40 | 60.08 | 53.33 |
| | Day 4 | 64.98 | 20.96 | 56.56 | 49.18 |
| | Day 5 | 66.01 | 19.36 | 56.95 | 50.10 |

Table 5: Similar to table 4 but for the winter period of 2021-22.

To shed more light on the model's performance in the simulation of AQI, we have drawn the performance diagrams (figure 6) for the model simulated AQI in different categories for both seasons, using the SR, Bias, CSI, and POD. The performance diagram (Roebber, 2009; Sengupta et al., 2022) provides a quick visualization of the model's performance for multiple statistical parameters. The category-wise statistical parameters have been plotted for Day 1 through Day 5 forecasts for post-monsoon (fig.6a) and winter (fig. 6b) seasons. In the performance diagram, an ideal model simulation would fall in the upper right corner. It is to be noted that the ideal value of Bias is 1, which indicates that the POD and SR match each other (Roebber, 2009; Sengupta et al., 2022). This signifies that the probability of getting a false forecast for a non-event from the model is equal to that of a false forecast for

an event from the same model. For the post-monsoonal period, the forecasts for very poor and above AQI fall relatively closer to the upper right corner, with POD values going up to 70% and SR reaching 100%. The model is highly (moderately) skillful in capturing the 'very poor and above' (moderate) air quality conditions. It depicts lower SR values (and thus higher FAR and Bias) for the poor AQI conditions; this is likely to be related to the underestimation of the very-poor AQI by the model resulting in higher occurrences of the simulated AQI in the poor category (in comparison with the observations), thus resulting in lower SR values for poor conditions, as noted in table 4.

For the winter season (fig. 6b), the model's performance shows large improvements, especially for poor AQI conditions (as noted in table 5). The POD and SR for 'very poor and above' conditions cross the 80% mark, indicating an excellent performance for Day 1 through Day 5 forecasts. Even for the poor category, the model shows large improvements with greater SR (~40%) and POD (~60%) values compared to the post-monsoon. Interestingly, as noted in tables 4 and 5, for both seasons, the model shows the highest performance ratings for the very poor and above AQI conditions. The implications of this have already been discussed in the analysis of tables 4 and 5. It is to be noted that, throughout section 3.1, we do not evaluate the model's performance for good (AQI ≤ 50) and satisfactory ($50 \leq AQI < 100$) categories as the observed AQI hardly ever falls in these categories. Nonetheless, the ability of the model to capture AQI in very poor and above conditions is encouraging as the air quality forecasting capabilities are mainly needed for such air quality conditions and not when the air quality is in a good or satisfactory category.

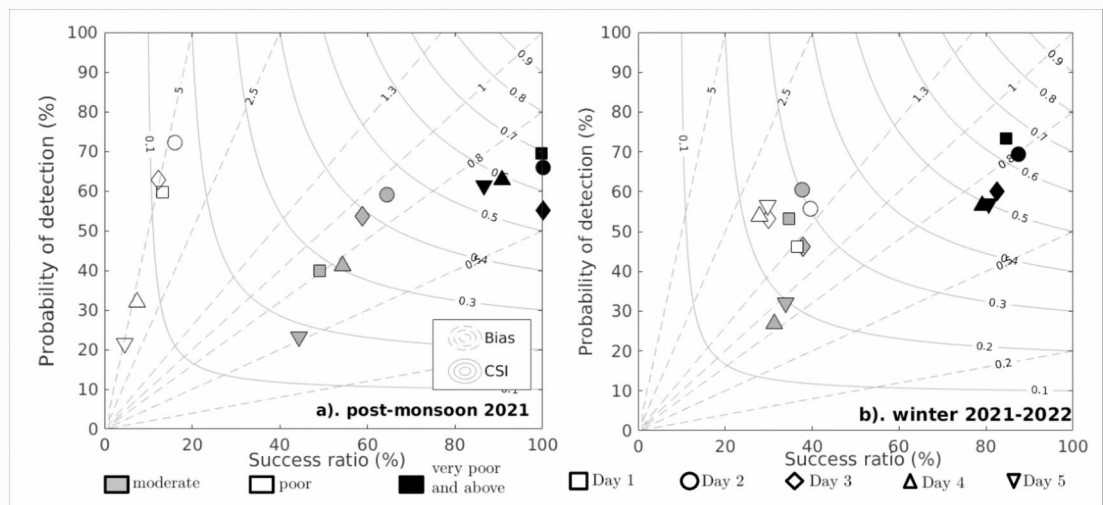

Figure 6. Performance diagram for model simulations of Air Quality Index for a) post-monsoon season 2021 and b) winter season 2021-22. The details about the calculation of the statistical parameters like Bias, and CSI can be found in section 5 of the supplementary material.

**3.2 Region and sector-wise source apportionment of PM$_{2.5}$ in Delhi:**

One of the main features of DSS is its ability to quantify the contribution of the different NCR districts and emissions sources to the PM$_{2.5}$ pollution load in Delhi. The tagged tracers employed in the system help achieve this objective. To facilitate ease in visualization and understanding of these contributions we divide them in six broad categories as follows a) Delhi transport sector, b) All other emission sectors within Delhi, c) Bordering districts (which include Jhajjar, Faridabad, Gurgaon, Gautam Buddha Nagar, Rohtak, Sonipat, Bagpat and Ghaziabad districts of NCR), d) Other districts of NCR (which include the remaining districts of NCR, the details of which can be found from figure 3) e) stubble

burning , and f). all other remaining regions. In figure 7, we show the daily mean and seasonal mean contribution of those six broad source-categories to the simulated $PM_{2.5}$ in Delhi for the post-monsoon and winter seasons of the year 2021. For the post-monsoonal period (fig. 7a-b), 34% contribution to $PM_{2.5}$ in Delhi comes from Delhi's own sources, including the transport, peripheral industries, residential, construction, waste burning, road dust, and energy sectors. The next major contribution comes from the bordering districts and the stubble-burning activities, with their seasonal mean contributions going up to 25% and 8% respectively. The stubble/biomass-burning activities impact the pollution load in Delhi roughly for a month i.e., from mid-October to mid-November. The daily mean biomass-burning contribution goes as high as 37% in the first week of November when the biomass-burning activities in Punjab and Haryana are recorded to be at their peak (Govardhan et al., 2022). It is important to note that around 26% of Delhi's $PM_{2.5}$ comes from the other regions (excluding the biomass burning activities), which are not included in the 20 districts considered in this analysis. Within Delhi, the major contribution comes from the transport sector with a seasonal mean of 17%.

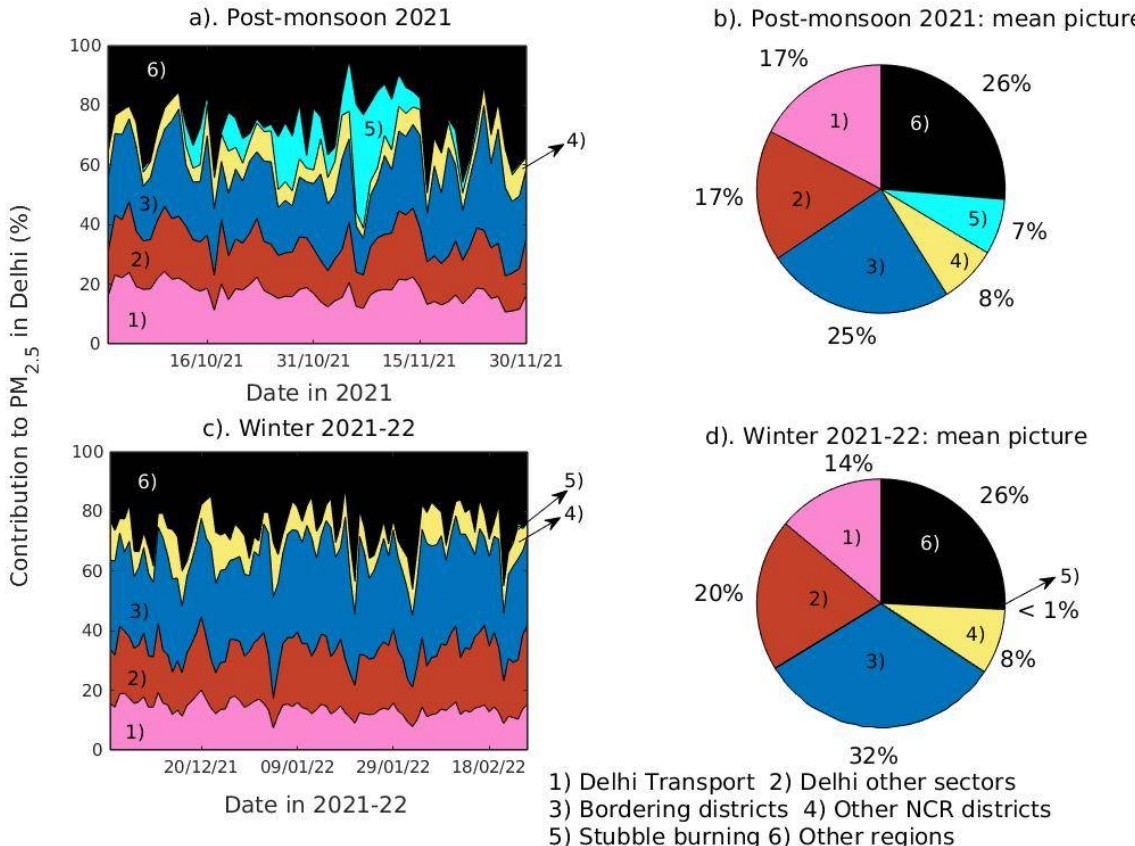

Figure 7: Source apportionment of $PM_{2.5}$ mass concentration in Delhi for a) post-monsoon 2021 on a daily mean basis b) post-monsoon 2021 on a seasonal mean basis c) winter 2021-22 on a daily mean basis, and d) winter 2022 on a seasonal mean basis. The numbers written on the pie charts indicate the percentage contribution of the particular source to $PM_{2.5}$ in Delhi. Day 1 forecasts have been used to generate this figure.

During the winter season (fig. 7c-d), Delhi's own contribution roughly remains the same (34%).

This estimate is comparable to a previous study carried out by TERI and ARAI, which reports the contribution to be around 36% (TERI and ARAI, 2018). The contribution from the neighboring districts increases to 20% from 17% in the post-monsoon season. Within Delhi, the transport sector contributes the highest (14%). The industries in and around Delhi also contribute around 9.5%. The increased contribution of the industries could be associated with the emissions coming from the brick kilns located on the periphery of the city. The kilns are not operational during the post-monsoon season, but they become operational during the winter season (TERI, 2018). The contribution from the 'other' regions remains roughly the same (26%) as in the post-monsoon season. Overall, on the seasonal mean basis, for the post-monsoonal season (winter season), contributions from the different regions could be listed as follows: Delhi: 34.4% (33.4%), NCR districts: 33% (40.2%), Biomass burning 7.3% (~0.1%) and the other regions: 27.3% (26.4%). Those bordering districts of Delhi contribute to around 25% in the post-monsoon season and 32% in the winter season. Thus a majority of the $PM_{2.5}$ in Delhi comes from its immediate neighbors. Thus, Delhi's air pollution load does not look like a local issue, but it seems to be a regional issue, and cooperation among various stakeholders is required to address this problem effectively.

**3.3 Impacts of emission reductions:**

The most unique feature of DSS is the availability of 'scenario' tracers. This feature estimates the impacts of reduction in the individual source/district-wise emissions on the $PM_{2.5}$ load in Delhi. We include 50 such $PM_{2.5}$ tracers, which carry the reduced emissions from 25 different sources, including 19 surrounding districts and the six individual emission sectors (namely transport, peripheral industries, waste burning, construction, road dust, and energy) in Delhi. We form two sets of scenario tracers with a) emissions reduced by 20% and b) emissions reduced by 40%. Using the scenario tracers one can compute the changes in the $PM_{2.5}$ mass concentration in Delhi upon a 20 or 40% reduction in one or a combination of the 25 emissions sources (19 surrounding districts and six sectors in Delhi). The reduction in $PM_{2.5}$ mass concentration in Delhi upon a 20% and 40% reduction in all those emissions during the post-monsoon and the winter seasons of 2021 have been plotted in figure 8. Similar to figure 7, we have divided the sources in four different categories i.e. the categories a to d from section 3.2.

During the post-monsoon season, a 20% reduction in all the sources (fig.8a) results in a seasonal mean reduction of ~12.1% in $PM_{2.5}$ Delhi. While around 5.7% of it would result from a 20% reduction in the sources within Delhi, the remaining 6.4% would come from the reduction in the neighboring districts of NCR. Similarly, a 40% reduction in all the concerned emissions sources (fig.8c) would result in an overall 24.3% reduction in the seasonal mean $PM_{2.5}$ load in Delhi, of which 11.5% comes from the reduction in the sources within Delhi, while the remaining 12.8% would result from 40% reduction in the emissions from other districts of NCR. It is to be noted that the change in $PM_{2.5}$ in Delhi roughly scales linearly from a 20% reduction to a 40% reduction. During the period when biomass burning activities are the highest (on 6th and 7th November 2021), the 20% (40%) reduction in other sources of $PM_{2.5}$ reduces the $PM_{2.5}$ in Delhi only by 7-8% (14-16%). Thus, it is to be noted that when such activities are at their peak, any control measure on the anthropogenic emissions of $PM_{2.5}$ will not have a drastic effect on the air quality in Delhi.

For the winter season, the 20% reduction scenarios result in a mean reduction of 13.8% in $PM_{2.5}$ in Delhi, of which 5.8% comes from Delhi's sources while the remaining 8% comes from the neighboring districts of NCR. Similarly, the 40% reduction scenarios result in a mean reduction of 27.75% in $PM_{2.5}$ in Delhi. Out of this, 11.5% comes from Delhi's own sources, while the remaining 16.25% comes from the other districts of NCR. In the winter season, the improvements in Delhi's $PM_{2.5}$ by controlling the emissions in the neighboring district of Jhajjar (see supplementary figure 4) are comparable to the improvements achieved by controlling the transport sector emissions within Delhi. However, in the post-monsoon season, the emission reductions in Jhajjar have a relatively lesser impact. This signifies the need for change in the emission reduction strategy from season to season for air quality management in Delhi.

The same policy for both seasons may not give the same amount of reductions.

On a daily mean basis, the reduction scenarios can reduce the PM$_{2.5}$ in Delhi by as high as 16% (for 20% reduction scenarios) and 32% (for 40% reduction scenarios) in either of the seasons. These control measures, when operated during severe air pollution events like the ones noticed during the last week of December 2021, the first week of January 2022, and the third week of January 2022, would result in a substantial reduction in Delhi's PM$_{2.5}$. The measurements of daily mean PM$_{2.5}$ suggest that the maximum values of PM$_{2.5}$ during those events were 334 μg m$^{-3}$, 310 μg m$^{-3}$, and 362 μg m$^{-3}$, respectively. The 40% reduction scenario for all the sources would result in ~25-30% reduction in PM$_{2.5}$ in Delhi during those days, which would roughly result in a reduction of 80-110 μg m$^{-3}$ in PM$_{2.5}$ in Delhi on those days. This would result in the modulation of air quality from the 'severe' category to the 'very poor' category. This is a satisfactory gain considering the already elevated air pollution level in the city. Thus, such information about the possible emission-reduction scenario would be critical from the air quality management perspective. Moreover, since the performance of the DSS in capturing the broad category of air quality scenario does not drastically drop from Day 1 to Day 5 (figure 5, table 2–5), such information would certainly help the decision-makers in managing the air quality in the city in a timely manner.

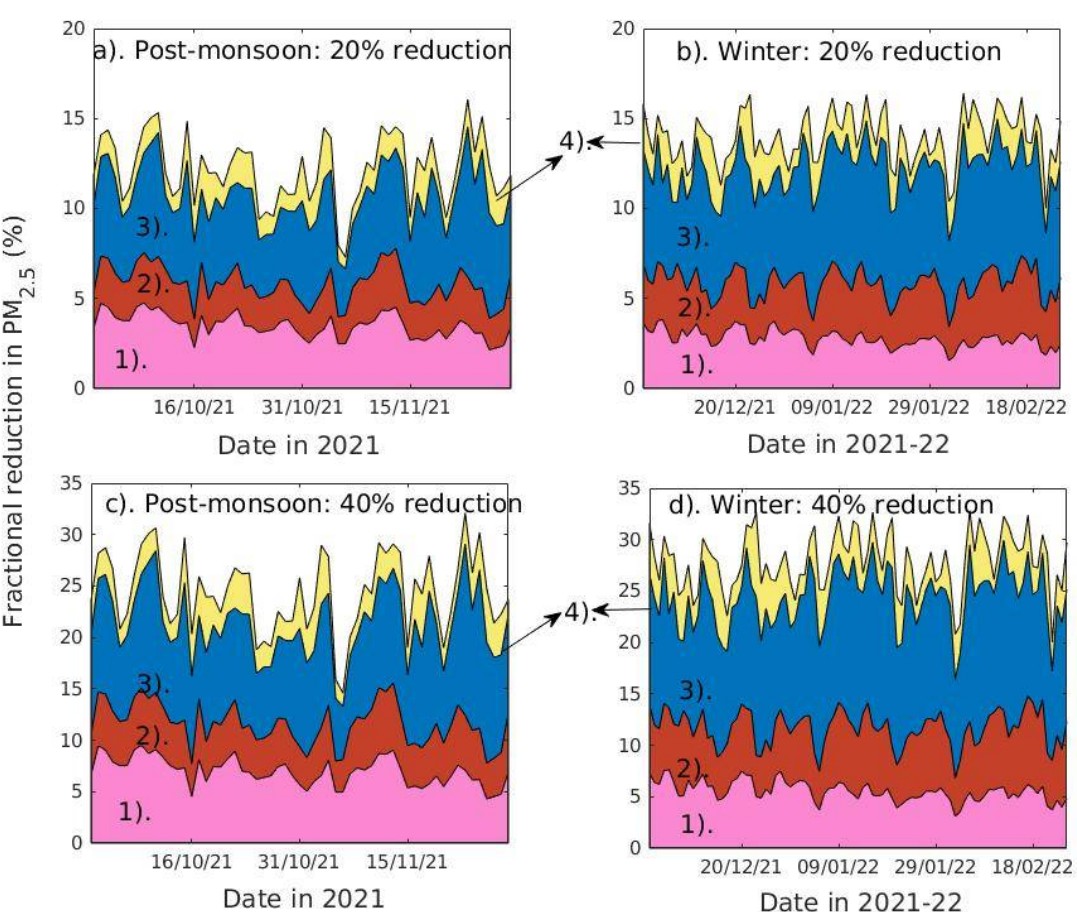

1). Delhi Transport  2). Delhi other sectors 3). Bordering districts  4). Other NCR districts

Figure 8: Fractional reduction in the PM$_{2.5}$ load in Delhi due to a). a 20% reduction in all the considered emission sources for the post-monsoon season of 2021, b) same as a) but for the winter

season, c). same as a) but with a 40% reduction scenario, and d) same as c) but for the winter season. Day 1 forecasts have been used to generate this figure.

        A practical example of the use of DSS for air quality management purposes in Delhi was witnessed in the month of November 2021. Based on the air quality forecast, source attribution of $PM_{2.5}$ in
Delhi, and the associated scenario analysis, the CAQM and the Government of Delhi issued certain restrictions on trans-boundary and internal vehicular traffic and construction activities in Delhi. This resulted in an 18-20% reduction in $PM_{2.5}$ and a 20-22% reduction in the AQI of Delhi (Ghude et al., 2022). This clearly signifies the role DSS played (and would play in the future) in the short-term air quality management in Delhi. This is one of the rare air quality forecasting systems in the world that offer
a utility like the 'scenarios' tool that would inform the decision-makers about the efficacy of their source-level interventions on the air pollution occurring in a city. With the help of the 'scenario' tool, users can create their own strategy for emission reduction to get an idea of how to possibly avoid the forecasted severe air pollution event for the city. We certainly note that DSS currently provides all such information only for the city of Delhi; however, there is an equal demand for such information from the neighboring
towns of NCR like Ghaziabad, Faridabad, Noida, Gurgaon, etc, as outlined in the recently formed air pollution control policy for the NCR (CAQM, 2022). In the next version of DSS, we plan to cater to this requirement and explore machine learning-based approaches to maximize computational efficiency. In the current configuration, the DSS runs with a relatively coarser resolution (10 km x 10 km). This is mainly due to the computational cost it carries associated with a large number of three-dimensional tagged tracers
and the upper bound on the daily run time due to the daily forecasting requirements. Nevertheless, in the next version, we are planning to increase the spatial resolution of the simulations. Another artifact of the coarse spatial resolution is the limited accuracy of the forecasts with respect to the observed $PM_{2.5}$ values. However, in the case of DSS, one is more interested in the relative contributions of the sources to the $PM_{2.5}$ load and the relative reduction in the $PM_{2.5}$ upon employing the various emission reduction
scenarios. This focus on the relative contribution comes from the basic assumption that the contributions would roughly remain similar even when the DSS-simulated $PM_{2.5}$ matches the observations with a greater agreement. However, we acknowledge that if the models underestimates the absolute concentration of $PM_{2.5}$, it is likely to erroneous source apportionment. Especially, during a severe air quality episode in the winter season, the contribution from the local sources would be much higher owing
to the stable atmospheric conditions. In such a situation, if the model fails to capture the peak, it will certainly underestimate the contribution from the local sources and overestimate the contribution coming from the relatively distant sources. We agree that the source apportionment in that situation would not be correct. However, we would also like to mention that, in situations where the model has missed the observational peaks, the modeled attribution for the local sources would represent a lower bound than the
reality. Thus, any intervention, if applied to the local sources, will certainly result in an enhanced reduction in $PM_{2.5}$ in the city in reality than that suggested by the model. Thus, in other words, in such situations, the modeled source attributions and the scenario analysis would represent a lower bound.
        Another reason for the underestimated $PM_{2.5}$ in DSS is the static nature of the emissions inventory. However, the anthropogenic emission sources vary in a dynamic manner. Any forecasting
model which does not take those dynamic changes into account is expected to miss the sudden rise in $PM_{2.5}$ associated with the dynamic changes in the emissions. Even though the chemical data assimilation operational in DSS bridges this gap at the start of the model run, it fails to capture the sudden rise in emissions happening during the other hours. The incursion of the dynamic emissions inventory, though, remains a challenge; there are a few recent efforts done on that front (Liu et al., 2018; Zhang et al.,2019;
Meng et al., 2020; Li et al., 2021). Using the daily Visible Infrared Imaging Radiometer Suite (VIIRS) thermal anomaly product, Zhang et al. (2019) and Li et al. (2021) have shown the capabilities of generating dynamic emissions for industrial sources. Meng et al. (2020) have utilized web-based traffic maps and real-time traffic data to generate a dynamic inventory of vehicular traffic emissions in China.

Such techniques could be used in future versions of DSS to get better estimates of real-time traffic. The emissions inventory used in this version of DSS does not take into account emissions associated with space heating. These emissions would be non-negligible, especially in the winter months. Thus, in the next version, we would explore the possibility of including such sources of emissions.

Additionally, we do acknowledge that the model's chemistry currently lacks the representation of the secondary aerosols in the ambient. There are several studies which have focused on understanding the 825 chemical composition of $PM_{2.5}$ in Delhi (Sharma et al., 2016; Sharma and Mondal, 2017; Jain et al., 2020; Yadav et al., 2022). A study by Sharma and Mondal, 2017 reports that, the particulate organic matter, soil/crustal matter, ammonium sulphate, ammonium nitrate, sea-salt and light absorbing carbon contribute 27.5%, 16.1%, 16.1%, 13.1%, 17.1% and 10.2% respectively to the city's $PM_{2.5}$. The study was carried out for a period of 2 years (January 2013 to December 2014). Jain et al., 2020 reported the chemical 830 composition of $PM_{2.5}$ for the period of 4 years (January 2013 to December 2016). The average $PM_{2.5}$ mass concentration for post-monsoon (winter) season was 186 (183) ug m$^{-3}$, out of which sulphates were reported to be 18.1 (18.6) ug m$^{-3}$, nitrates were 18.4 (20.2) ug m$^{-3}$, chlorides and ammonium were 11.4 (11) and 14.9 (16.6) ug m$^{-3}$ respectively. The elemental carbon and organic carbon were measured to 11.4 (10.6) and 25.2 (23.6) ug m$^{-3}$ respectively. Thus, it may be seen that the missing aerosol species (mainly 835 the nitrates, ammonium and chlorides) in the GOCART mechanism of WRF-Chem contribute to around 24-30% of $PM_{2.5}$ in Delhi. Thus a part of the underestimation in the model could be associated with these missing species. An artefact of the missing secondary aerosols in the model is that the tracers are mainly put on the primary species BC and OC. However, the secondary species are not tagged effectively. This results in underestimated impacts of the source levels interventions on the ambient $PM_{2.5}$. For example, in 840 reality, the traffic emission reductions might lead to a significant reduction in nitrate aerosols, but this is not captured by the model. Thus, the model currently underestimates the impacts of source-level interventions. In the next version, we are aiming to include the missing secondary aerosols by using a simple parameterization (Hodzic and Jimenez, 2011). This would include nitrate and secondary organic aerosols in the model set-up without hampering the model runtime drastically.

The biomass-burning emissions, on the other hand, have even more uncertainties. The limitations associated with satellite detection of stubble-burning fires due to the cloud cover (Liu et al., 2020; Cusworth et al., 2018), the limited number of passes in a day (Liu et al., 2020; Kumar et al., 2021), smarter burning practices (Kumar et al., 2021), unrealistic estimation of emissions from the fires (Kumar et al., 2021), etc., lead to multiple orders of uncertainty in the emission estimates from fires. We have seen 850 that biomass-burning fires contribute as high as 37% to the daily mean $PM_{2.5}$ load in Delhi during the peak burning periods; however, this number certainly represents a lower bound due to the aforementioned uncertainties. Therefore, more work is needed to constrain the estimates of the emissions from biomass-burning activities in the region. Additionally, stronger policies are needed to reduce the amount of stubble that is being burnt, especially in the post-monsoonal season in this region. In DSS, we carry out the 855 chemical data assimilation only once in the forecasting cycle in this setup; in the future, we can carry out assimilation at least twice to correct the model concentrations even at night times. This will help the model capture higher PM concentrations which usually occur during the night hours due to shallower mixed layers. In the next version of DSS, we are planning to incorporate a few new scenario tracers, like the 'odd-even' scenario for vehicular traffic, which allows only those vehicles to ply on the road with an 860 odd (even) number as the last digit of their registration number on odd (even) dates. This policy has been used by the Government of Delhi in the past to control vehicular movement and the associated emissions (Sud and Iyengar, 2016; Kumar et al., 2017; Choudhary et al., 2018; Tiwari et al., 2018). Thus, while the first version of DSS has proven to be beneficial for the policy-makers, we have identified its limitations as well, and we will attempt to overcome those limitations in the next version.

Additionally, we understand the the day-4 and day-5 forecasts would be more useful for the policy makers. We acknowledge that the implementation of source-level intervention may not start from day-1, so in reality, one also needs to account for a time delay in implementing those. This is currently

missing in the framework. However, we would also like to mention that including such time-delays, even for the interventions only on the major sources, if not all, would substantially increase the number of tracers in the modeling framework. Currently, we have more than 400 three-dimensional tracers in an operational forecasting set-up. Keeping in mind our operational commitments, we will certainly include some sense of the time delay for the scenario tracers in the next version of DSS.

**4. Conclusions**:

In order to assist the governing authorities in managing the air quality in the capital of India, Delhi, we have designed an operational air quality forecasting framework with certain unique features that help the decision-makers to form policies for managing the air quality in the city. This newly developed Decision Support System (DSS) for air quality management in Delhi, besides forecasting the air quality in the north Indian region for the next five days, quantifies the contributions of the 19 surrounding districts, individual emission sectors in Delhi, and the biomass burning activities (occurring primarily in the northwestern states of India in the post-monsoon season) to the $PM_{2.5}$ mass concentration in Delhi. The system also quantifies the effects of emission source-level interventions on the forecasted air pollution in the city. Thus, with the help of DSS, the policy-makers not only get a warning about future severe air pollution events but also understand the possible causes for the event and get a quantitative idea about the efficacy of the source-level interventions on the forecasted event. In this paper, we evaluated the performance of DSS in simulating near-surface $PM_{2.5}$ mass concentration and the associated air quality index in Delhi for the post-monsoon and winter seasons of 2021-22. We also carry out the source apportionment of $PM_{2.5}$ in Delhi during the two seasons. The key results are listed as follows:

1. The performance of the model in simulating the air pollution in Delhi noticeably improves from post-monsoon to the winter season, owing primarily to the uncertainty in the emission estimates from the biomass burning activities and the anthropogenic activities during the Diwali festival, which occur in the post-monsoon season.

2. For both seasons (post-monsoon and winter), the DSS satisfactorily captures the observed air quality index (AQI) in Delhi, especially when the AQI crosses a very poor or above that mark. Under such a situation, DSS depicts a very low false alarm ratio (~20%), which increases the trustworthiness of the simulated AQI. For all the AQI categories (moderate, poor, and very poor and above), DSS shows a very high accuracy (~80%). However, the critical success index for the simulated AQI is seen to be the highest for the 'very-poor and above' category, i.e., extreme pollution events are captured very well.

3. The performance of the model does not deviate largely from Day 1 to Day 5 forecasts, which highlights the applicability of the DSS forecasts in short to medium-range air quality management activities.

4. The region-wise source apportionment of $PM_{2.5}$ mass concentration in Delhi carried out with the help of DSS suggests that during the post-monsoon season (winter season), on average, Delhi itself contributes 34.4% (33.4%) to its $PM_{2.5}$ load. The NCR districts contribute 31% (40.2%). The emissions from the biomass burning activities on the seasonal mean basis contribute 7.3% (~0.1%) of the $PM_{2.5}$ mass in Delhi, while the other regions contribute around 27.3% (26.4%). The districts of NCR which share their border with Delhi (namely Jhajjar, Gurgaon, Faridabad, Ghaziabad, Gautam Buddha Nagar, Bagpat, and Sonipat) contribute about 22% in the post-monsoon season and 30% in the winter season.

5. The 'scenario' tracers employed for $PM_{2.5}$ in DSS suggest that a 20% reduction in all the tagged sources in Delhi and the NCR districts results in a seasonal mean reduction of ~12 - 14 % in $PM_{2.5}$ mass in Delhi. While around 5.8% of that comes from controlling Delhi's own emission sources, the remaining comes from control measures applied in the NCR districts. As expected, during the peak biomass burning events, such control measures on the anthropogenic emissions yield a relatively lesser gain.

6. The reduction in Delhi's $PM_{2.5}$ load scales roughly linearly with the magnitude of emission reductions, i.e., the reduction in Delhi's $PM_{2.5}$ for a 40% control on the anthropogenic emission sources within Delhi

and the NCR districts is roughly twice that of the reductions associated with a 20% cut on emissions.
    In short, DSS is a highly effective tool for decision-makers and the masses.

**Author contribution:**
Design and Model development: GG, SG, RK, PG and CJ
Input dataset: SS
Analysis and interpretation of the result: GG, SG and RK
Figures: GG and SI
Writing original draft: GG
Modifying the draft: All authors
Project supervision: SG, RK, RN and MR

**Acknowledgments:**
The authors would like to thank the Central Pollution Control Board (CPCB) and the Delhi Pollution
Control Committee (DPCC) for providing measurement data for $PM_{2.5}$ which was utilized in this study for chemical data assimilation and model evaluation purpose. The authors also thank the MODIS team for providing satellite retrieved AOD and VIIRS team active fire count information. The study was partially supported by Commission for Air Quality Managament in the National Capital Region and its adjoining areas, and, National Supercomputing Mission. The authors would like to thank Director, IITM for his
constant support and motivation. The simulations were carried out on 'Aditya' supercomputer at IITM. The National Center for Atmospheric Research is sponsored by the National Science Foundation.

**Code/data availability Statement:**
The observational data for $PM_{2.5}$ from Central Pollution Control Board, India, is available on
https://app.cpcbccr.com/ccr/#/caaqm-dashboard-all/caaqm-landing. The model output is available on IITM HPC and could be downloaded through a request to the corresponding authors. The model code employed in DSS (Govardhan, 2022) is available at  https://figshare.com/articles/software/WRF-Chem-DSS-Model/21655883. The user manual (Govardhan, 2023) for installing and running the code is available at https://figshare.com/articles/software/User_manual_for_DSS/22231543.
**Conflict of Interests:**
The authors declare that they have no conflict of interests.

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
