# Peer review of "Decision Support System version 1.0 (DSS v1.0) for air quality management in Delhi, India."

_Geoscientific Model Development, 2022_

## Author Comment (AC1)

**Decision Support System (v 1.0) for Air Quality Management in New Delhi, India**

Gaurav Govardhan[1,2,*], Sachin D. Ghude[1,*], Rajesh Kumar[3], Sumit Sharma[4], Preeti Gunwani[5], Chinmay Jena[5], Shubhangi Ingle[1], Prafull Yadav[1], Sreyashi Debnath[1,6], Pooja Pawar[1], Prodip Acharja[1,6], Rajmal Jat[1], Gayatry Kalita[1], Rupal Ambulkar[1], Santosh Kulkarni[7], Akshara Kaginalkar[7], Vijay K Soni[5], Ravi S. Nanjundiah[8,9], and M. Rajeevan[10]

1: Indian Institute of Tropical Meteorology, Ministry of Earth Sciences, Pune, Maharashtra, India
2: National Centre for Medium-Range Weather Forecasting, Ministry of Earth Sciences, Noida, Uttar Pradesh, India
3: National Center for Atmospheric Research, Boulder, CO, United States of America
4: The Energy and Resources Institute, Delhi, India
5: India Meteorology Department, Ministry of Earth Sciences, Delhi, India
6: Savitribai Phule Pune University, Pune, Maharashtra, India
7: Centre for Development of Advanced Computing, Pune, Maharashtra, India
8: Centre for Atmospheric and Oceanic Sciences, Indian Institute of Science, Bengaluru, India.
9: Divecha Centre for Climate Change, Indian Institute of Science, Bengaluru, India.
10: National Centre for Earth Science Studies, Thiruvananthapuram, Kerala, India.
* : corresponding authors:
 Gaurav Govardhan: gaurav.govardhan@tropmet.res.in ,
Sachin D. Ghude: sachinghude@tropmet.res.in

**Replies to the reviewers**

We thank the reviewers for their insightful comments on the manuscript. We believe that the comments/suggestions have certainly improved the state of the manuscript. In this document, we have provided point by point reply to the comments/suggestion and will modify the manuscript wherever required. The reviewer's comments are written in black font, and the authors' replies are written in blue font.

**Comments from Reviewer 1**

Govardhan et al. introduce DSS1.0, an operational air quality forecasting and emission scenario framework with high resolution WRF-Chem simulations at its core along with other enhancements such as chemical data assimilation from ground-based stations and satellites to improve initial conditions and tagged tracers for different sectors and regions with different emission levels. Their forecast system has the novelty of providing real-time source apportionment of PM2.5 along with the forecasts as well as sufficient tagged tracers from different sectors and regions to build bespoke short-term episode-specific mitigation policy. This is commendable work and should be very attractive for policymakers.

The paper is well-written with clearly stated objectives. There are also some notable details

which show the scientific robustness of the work, for example, the details of the chemical data assimilation as applied to the tagged tracers, and the "feedback" module in the forecasting system which adjusts the new initial conditions after an actual policy is implemented in the real world.

However, I'd like to point out some weaknesses which need to be addressed before the work is published:

1. The authors explained the underestimation of PM2.5 concentrations in the first half of November due to the lack of firecracker emissions in the model during the Diwali festival, which is understandable, but they haven't explained the dramatic overestimation around the third week of November (even if the policy intervention was not included in the model). Is this due to wrongly generated timing of fire emissions or a poor meteorological forecast during that week? This needs to be discussed in sufficient detail.

- We thank the reviewer for raising this critical point. As mentioned in line number 415-421 of the original manuscript, we believe that the main reason behind the mismatch between the model and observations is due to the missing policy interventions in the model framework. We further note that the fire activity in the neighboring states of Punjab and Haryana during that period was also on a declining trend (Fig.1, Govardhan et al., 2023), so the associated fire emissions may not be completely responsible for this behavior of the model. Though we agree that the fires had not completely ceased by then, so there is a possibility of fire emissions being wrongly carried to Delhi in the model's atmosphere, unlike the observations. While carefully carrying out the analysis suggested by the reviewer, we have identified one issue with the model simulations carried out for that time period. The prescribed anthropogenic emissions for that period seem to be erroneous. They appear to have deviated from their original values, possibly due to some runtime errors. Due to such erroneous emissions, the associated simulated meteorological parameters would also be affected. So comparing the erroneous meteorological fields with the corresponding observations would not reveal the correct picture. So, we have decided to redo the model simulations for those 7 days. We will correct the erroneous prescription of anthropogenic emissions and then examine how the model performs during that period. We will further analyze the simulated meteorological fields in comparison with observations. The simulations have already started. However, carrying out those in forecast mode with all the necessary data to be ingested in them is a challenging task. We are currently working on that. We will include this analysis in the modified version of the manuscript. We believe that these new simulations though, would modify the model-observation comparison for those 7 days, would not drastically impact the overall conclusions of this study. We further believe that even after correcting for emissions, though the model's performance would improve, it would still overestimate in comparison with observation owing to the absence of policy interventions in the simulations. As suggested by the reviewer in comment #7, the emission reduction scenarios would help us understand the skill of the scenario tool of DSS in simulating the reduced ambient emissions. We thank the reviewer for suggesting us to examine this point. This analysis will certainly revise some of the conclusions made for that particular period of 7 days. We believe that this new analysis would make the paper stronger and more rigorous. In the revised manuscript, we will include an analysis of the factors causing the overestimation in modeled values.

2. The forecasts completely miss three large observed peaks in winter (Fig 5c). This surely

cannot be due to poor representation of biomass burning in the model because there's hardly any agricultural burning going on during this period in the region. The authors have touched upon the issue of lack of a dynamic emission inventory, but can these peaks be captured just by applying a temporal cycle to the existing emissions, or are some key emission sources/processes missing in the model? The authors need to discuss this in more detail. I'm thinking of open-waste burning, brick kilns, gas-to-particle conversion etc. but a bit more local knowledge needs to be added here.

- We agree with the reviewer. The current emissions inventory used in the model though does have some information about the sources like open waste burning and brick kilns, in and around Delhi, this information is likely to be underestimating the reality in 2021. The emissions inventory employed in this study was compiled using surveys done in 2016. There are significant changes that have occurred in the emissions magnitudes from 2016 to 2021. We agree that these uncertainties will affect the model simulations. Moreover, during the January month the temperatures region fall down. The residents of Delhi burn biomass or solid wood for space heating purposes. Such sources are missing in the employed emissions inventory. Additionally, such burning activities occur at a very fine spatial scales which can not be identified by remote sensing techniques. Thus, a part of the underestimation during the month of January would be related to these factors. In addition to this, the lower temperatures bring foggy conditions into the picture. Such weather conditions promote a large number of atmospheric chemical reactions resulting in gas-to-particle conversion of volatile gas phase species into secondary aerosols. Such processes are currently missing the models' chemical mechanism. This further enhances the underestimations in the model. All these factors put together result in the underestimated $PM_{2.5}$ in the model vis-a-vis the measurements.

As suggested by the reviewer, we will include this discussion in the revised version of the manuscript.

3. The defence that the source apportionment results shall hold true on a percentage basin is even when the model underestimates the episodes is only partially true. This is because, the relative contributions of local and near-regional sources might disproportionately increase during the the peaks which will not be reflected in the source apportionment results if those peaks are not captured. This is of utmost importance because these are exactly the periods when a policy implementation might be wanted. Therefore, the authors need to acknowledge and discuss this weakness in the forecast system and propose potential solutions.

- We agree with the reviewer. Especially, during a severe air quality episode in the winter season, the contribution from the local sources would be much higher owing to the stable atmospheric conditions. In such a situation, if the model fails to capture the peak, it will certainly underestimate the contribution from the local sources and overestimate the contribution coming from the relatively distant sources. We agree that the source apportionment in that situation would not be correct. The possible solutions to tackle this issue would include a) improved horizontal resolution of the model b). Improved representation of aerosols in the model c). Inclusion of secondary aerosols in the model's chemical mechanism d). Modifications in the employed emissions inventory in the modeling framework e). Better representation of biomass burning related emissions in the model f) corrected meteorological forcing for the simulations and many other. We will discuss this in detail in the revised version of the manuscript. However,

we would also like to mention that, in situations where the model has missed the observational peaks, the modeled attribution for the local sources would represent a lower bound than the reality. Thus, any intervention, if applied to the local sources, will certainly result in an enhanced reduction in $PM_{2.5}$ in the city in reality than that suggested by the model. Thus, in other words, in such situations, the modeled source attributions and the scenario analysis would represent a lower bound. We will also discuss this point in the revised version of the manuscript.

4. The aerosol module used in WRF-Chem doesn't represent secondary organic and inorganic aerosol production. This is understandable and the authors have defended their choice well given the computational constraints of running a nested high-resolution model along with several tracers especially when PM2.5 isn't directly simulated, and they had to tag various PM-components for each sector and region. However, lack of gas-to-particle conversion in the model will have an impact on the contributions in the scenarios. For example, in the real world, traffic emission reductions might lead to a significant reduction in nitrate aerosols, but this won't happen in the model. The same goes for energy/industry emissions and sulfate aerosols. In that sense, the air quality improvements from the scenarios in the DSS might be underestimated – this needs to be clearly discussed.

- We thank the reviewer for this important suggestion. We do acknowledge that the model's chemistry currently lacks the representation of the secondary aerosols in the ambient. This will certainly limit the model's ability to simulate the observed aerosol field correctly. However, as mentioned by the reviewer the absence of the secondary aerosols would also result in underestimated impacts of the source levels interventions on the ambient $PM_{2.5}$. We agree with the reviewer. We will discuss this point in detail in the revised version of the manuscript. Moreover, we would like to mention that in the next version of DSS we are aiming to include the missing secondary aerosols by using a simple parameterization (Hodzic and Jimenez, 2011). This would include nitrate and secondary organic aerosols in the model set-up without hampering the model runtime drastically. This will also be further discussed in the revised version of the manuscript.

5. For Figures 7 and 8, which of the forecast has been used: 1-day, 2-day, 3-day, 4-day or 5-day forecast? This needs to be stated in the figures.

- We thank the reviewer for raising this important issue. We have used the day-1 forecasts for the analysis of the scenarios in Figures 7 and 8. We will mention this explicitly in the revised manuscript.

6. When suggesting the policy recommendation based on scenarios, I suppose the 4-day or 5-day forecast will be more practical than the 1-day forecast as it will allow some time for decision-making. However, the 4-day or 5-day forecasts will have a 20% or 40% reduction throughout those 4 or 5 days based on the tagged tracers, while the policy might be implemented at a later start date – this may lead to discrepancies in the outcomes. Therefore, the authors need to make this point clear.
- We agree with the reviewer. The day-4 and day-5 forecasts would be more useful for the policy makers. We agree with the reviewer that implementation of source-level intervention may not start from day-1, so in reality, one also needs to account for a time delay in implementing those. This is currently missing in the framework. We will certainly discuss this in the revised

manuscript. However, we would also like to mention that including such time-delays, even for the interventions only on the major sources, if not all, would substantially increase the number of tracers in the modeling framework. Currently, we have more than 400 three-dimensional tracers in an operational forecasting set-up. Keeping in mind our operational commitments, we will certainly include some sense of the time delay for the scenario tracers in the next version of DSS, however, we will try to keep it limited, given the computational load it carries. Nevertheless, we would like to thank the reviewer for raising this important point. We will explicitly discuss this in the revised version of the manuscript.

7. A proper evaluation of this system should include not just the forecast performance against observations but also the accuracy of scenarios. Therefore, the policy intervention that happened post-Diwali should ideally be evaluated against the closest possible scenario based on the tracers. Does the implementation of the closest possible emission scenario resembling the post-Diwali policy intervention successfully reproduce the drop in PM2.5 during that period? This would be a real litmus-test of the system, and even if it doesn't reproduce the drop accurately, we will at least learn about the discrepancies and get a step closer to making the modelling system reflect the real-world conditions. Such an evaluation will inform emission inventory modifications or new process representations. Therefore, it would be very valuable for the community if the authors can perform this evaluation.
- We thank the reviewer for this useful suggestion. This presents a rare opportunity for us to test the scenarios in the model. This will also help us understand the other factors responsible for the model's disagreement with the observations. It will also shed light on the possible discrepancies in the emissions inventory in the model. As mentioned in the reply to the reviewer's comment 1, we are freshly carrying out the simulations for that time period. In the revised version of the manuscript, we will certainly include this analysis.

Once again, I commend the authors for this crucial and significant work, and I have no hesitation in recommending it for publication in GMD once these issued are addressed.

- We thank the reviewer for the insightful comments. We believe that the manuscript will improve substantially upon including the suggestions and discussions pointed out the by the reviewer.

**Comments from Reviewer 2:**

The manuscript "Decision Support System version 1.0 (DSS v1.0) for air quality management in Delhi, India" by Govardhan et al. introduces a model system for short-term air quality forecast and emission reduction strategies in Delhi during the post-monsoon and winter seasons of 2021-2022. The authors use the WRF-Chem model with specific emission inventories to forecast the regional PM2.5 concentrations and Air Quality Index (AQI) over Delhi in five days, and they add some tagged-tracers to quantify the contributions of emissions from different sources over the local and surrounding regions of Delhi. The authors also design two scenarios of emission reductions to evaluate the impacts of sectorial anthropogenic emissions on the PM2.5 levels in Delhi.

In general, the manuscript is well organized and written, which fits the scope of the Geoscientific Model Development. The proposed model system in this study is promising to warn the short-term air pollution events and it's useful for local policymakers to manage the air quality in Delhi. However, it would be better if the authors can describe and discuss more details on how to estimate the biomass burning emission inventory, as well as its uncertainty and impact on forecasting PM2.5. It would be more convincing if the authors can provide some observations of PM2.5 compositions to evaluate the model simulations, as they use a relatively simple aerosol scheme lacking some secondary aerosol species. The authors might uniform and enlarge the labels and legends in figures for a better reading experience. The reviewer recommends publication after major revisions. Please see the specific comments and technical corrections listed below.

- We thank the reviewer for the constructive comments and suggestions on the manuscript. We think that including the discussions suggested by the reviewer would help the readers understand the manuscript better.  In our replies to the reviewer, we have thoroughly addressed the two important points suggested by the reviewer about the generation of fire emission fields and the comparison of the PM$_{2.5}$ composition in the model against the observations. In the revised version of the manuscript, we will improve the readability of the labels and legends in the figures and tables. In this document, we have provided point by point reply to the comments/suggestion and have modified the manuscript wherever required. The reviewer's comments are written in black font, and the authors' replies are written in blue font.

**Specific comments**

P3, Line 130: Which version of WRF-Chem do you use? Please add the citations for the WRF-Chem model.

- We have used version 3.9.1 of WRF-Chem. In the revised manuscript, we have added the corresponding citation.

P4-5, Line 150, 169, 193: Please add the citations for the IITM GFS, EDGARv4.3 inventory, and the MODIS active fire count data.

- We thank the reviewer for this suggestion. In the revised manuscript, we have added the corresponding citations.

P4, Line 179: The authors use the anthropogenic emission inventory from TERI for the year 2016. Is there any increasing or decreasing trends of anthropogenic emissions from 2016 to 2022?

- We thank the reviewer for raising this important issue. In general, for Delhi-NCR, there is an increasing trend in the anthropogenic emissions in the recent years. Sahu et al., 2023 reports changes in sectoral emissions over Delhi in 2020 in comparison with 2010. The study suggests that for $PM_{2.5}$, the emissions from transport sector and industries have increased by 37 and 25% respectively. On the other hand, the residential sector emissions show a slight decrease (1-2%). However, there is no such data for the period 2016 to 2022. Secondly, Sahu et al., 2023 compares two such inventories that are prepared a single group. However, there is a substantial variation in the estimates of Delhi's total $PM_{2.5}$ emissions and the corresponding sectoral contributions to it in the existing inventories prepared by multiple working groups as clearly outlined by Jalan and Dholakia, 2019. Thus, one can not simply use such ratios put forth by Sahu et al., 2023 and modify the emission in other inventories accordingly. Moreover, most of these inventories are not publicly available, thus implementing those in the models becomes a very challenging task. Keeping in mind, all these issues we have employed the TERI 2018 inventory in DSS with no scaling for 2021.

P5, Line 192: I would suggest the authors to give more information on the forecasted fire emissions. I'm wondering if this method can capture the day-to-day variability of fire emissions in a short forecast period based on the climatological fire emissions.

- We thank the reviewer for raising this important point. Following are the details about the methodology opted for generating real-time and near-future fire emissions.

We have prepared a daily climatology for year-long fire emissions using the Fire InveNtory from NCAR (FINN) data-set for the years 2002 to 2018. On each day of the forecast, we superimpose the near-real time daily active fire count data from the Visible Infrared Imaging Radiometer Suite (VIIRS) instrument on-board the Suomi National Polar-Orbiting Partnership (Suomi NPP) satellite on the climatological fire emissions file for that day. For day 1 of the forecast, the fire emissions only over those grids are activated where we get non-zero active fire counts on that day with a confidence level greater than 70%. The other points in the domain are supplied with no fire emissions. For day 2 – day 5 of the forecast, the climatological fire emissions over only those grids are activated where we get non-zero values in the climatological VIIRS fire count data for that day. This dataset is prepared using the VIIRS data the years 2011–2018. Thus, while the Day 1 fire emission forecasts are generated by amalgamation of near-real time fire count and climatological fire emissions, the Day 2-- Day 5 fire emission forecasts are generated using the climatological information about the fire emissions and the active fire counts.

We attempt to include some information about the real-time using the active fire count data. However, we understand that there is a large uncertainty associated with the fire count information too. Cusworth et al., 2018 has reported inconsistencies in the regional total Fire Radiative Power (FRP) due to clouds, haze, and smoke. The persistent dense haze reduces the brightness temperature over a region, thus interfering with the detection of thermal signatures of small fires by VIIRS. Another major problem is that the local farmers are getting better at avoiding fire detections by satellite. Fires are often started in the night hours or on days with

cloud cover to avoid detection by the satellites. High particulate matter levels and relatively low fire counts on the preceding days of rains suggest that high burning activity takes place on those days under clouds. Moreover, sometimes the stubble is collected over a small region of the field, and it is then set on fire to avoid detection by the footprint of the satellites. Another critical issue is associated with the temporal frequency of the pass of the satellites. The satellites pass over the region once a day in the morning to afternoon hours. So, fires occurring at all other times are missed by the satellites. Additionally, an underestimation of estimated stubble burning emissions from the north-western region of India has been linked to inability of detecting partially burnt areas (Liu et al., 2018). Thus, all the aforementioned limitations adversely affect satellites' detection of stubble-burning over Punjab and Haryana. These uncertainties would reflect into the uncertainties in the generated emission fields. We further acknowledge that it is very challenging to forecast the emissions happening from such anthropogenic agricultural fires. In this study we are employing the climatological fields of emissions for day 2 – day 5. However, in future we are aiming to improve on these assumptions. We are currently attempting deep learning techniques to generate such emissions for day 2 and day 3 of the forecasts are showing some promise (Gaikwad et al., 2023), which. In the next version of DSS, we plan to employ such techniques.

P6, Line 260: I'm wondering if lack of nitrate and ammonia aerosols in this model would cause the biases in total PM2.5 concentrations. It seems that Figure 2 is based on the WRF-Chem simulation. I would suggest the authors to show some observations of speciated PM2.5 concentrations or cite some previous studies.

- We agree with the reviewer. The absence of nitrate and ammonia aerosols in the model's aerosol scheme would certainly play an important role in the model underestimations on $PM_{2.5}$. However, as mentioned in the manuscript, for an operational DSS we stick to the simplest chemical mechanism keeping in mind the computational load and operational constraints. However, in the next version of DSS we are aiming to include the missing secondary aerosols by using a simple parameterization put forth in Hodzic and Jimenez, 2011. This would include nitrate and secondary organic aerosols in the model set-up without hampering the model runtime drastically.

The figure 2 in the manuscript is based on the model simulations. There are several studies which have focused on understanding the chemical composition of $PM_{2.5}$ in Delhi (Sharma et al., 2016; Sharma and Mondal, 2017; Jain et al., 2020; Yadav et al., 2022). A study by Sharma and Mondal, 2017 reports that, the particulate organic matter, soil/crustal matter, ammonium sulphate, ammonium nitrate, sea-salt and light absorbing carbon contribute 27.5%, 16.1%, 16.1%, 13.1%, 17.1% an 10.2% respectively to the city's $PM_{2.5}$. The study was carried out for a period of 2 years (January 2013 to December 2014). Jain et al., 2020 reported the chemical composition of $PM_{2.5}$ for the period of 4 years (January 2013 to December 2016). The average $PM_{2.5}$ mass concentration for post-monsoon (winter) season was 186 (183) ug/m3, out of which sulphates were reported to be 18.1 (18.6) ug/m3, nitrates were 18.4 (20.2) ug/m3, chlorides and ammonium were 11.4 (11) and 14.9 (16.6) ug/m3 respectively. The elemental carbon and organic carbon were measured to 11.4 (10.6) and 25.2 (23.6) ug/m3 respectively. Thus, it may be seen that the missing aerosol species (mainly the nitrates, ammonium and chlorides) in the GOCART mechanism of WRF-Chem contribute to around 24-30% of $PM_{2.5}$ in Delhi. Thus a part of the underestimation in the model could be associated with these missing species. In the next version

of DSS, we plan to include chlorides using a simple parameterization equation relating with to HCl  as a function of relative humidity (Pawar et al., 2023). Additionally, as mentioned above, we also plan to include nitrate and ammonium using a the scheme suggested by Hodzic and Jimenez, 2011.  In the revised manuscript we will include these points.

P11-12, Section 3.1: The authors explain the underestimation of PM2.5 concentrations in the first week of November is due to the large uncertainty of fire emissions. As I mentioned above, I would suggest the authors to discuss more about the method on predicting the daily fire emissions, because the daily variation of fire emissions may vary with the weather conditions and some human activities, which may introduce the large biases during the burning seasons. Could the authors show the estimated fire emissions during this period? I'm also wondering if the overestimation of PM2.5 concentrations in the following weeks is caused by the inaccurate anthropogenic emissions or the fire emissions.

- We thank the reviewer for raising this important concern. We would like to mention that the fires occurring in agriculture fields of the neighboring states of Punjab and Haryana are purely anthropogenic in nature. The paddy residue in the agriculture fields is cleared by setting on fire. This is done in order to clear the field for the next cropping season. Thus, weather does not play as much decisive role in these fires as it plays in forest fires. Nevertheless, we agree that the entire process of estimating the fire emission using active fire count data does depend on weather conditions mainly the cloud cover. In reply to the reviewer's comment for P5, Line 192, we have described the methodology employed to generate the fire emissions in the forecast. We have also described the uncertainties associated with the methodology.

The estimated fire emissions for organic carbon species during the period of 1st November 2021 to 7th November 2021 are shown in figures 1 – 7 below. It may be noted that climatologically the fires occur mainly in the northwestern states of Punjab during that period. The near-real-time fire count information from VIIRS decides which locations in the domain are to be supplied with the climatological fire emissions.

[Figure]

[Figure]

**Emissions Organic Carbon ($\mu$g m$^{-2}$ s$^{-1}$) from biomass burning on 2nd November 2021**

Figure 2

[Figure]

**Emissions Organic Carbon ($\mu$g m$^{-2}$ s$^{-1}$) from biomass burning on 3rd November 2021**

Figure 3

[Figure]

**Emissions Organic Carbon ($\mu$g m$^{-2}$ s$^{-1}$) from biomass burning on 4th November 2021**

Figure 4

[Figure]

**Emissions Organic Carbon ($\mu$g m$^{-2}$ s$^{-1}$) from biomass burning on 5th November 2021**

Figure 5

[Figure]

**Emissions Organic Carbon ($\mu$g m$^{-2}$ s$^{-1}$) from biomass burning on 6th November 2021**

Figure 6

[Figure]

**Emissions Organic Carbon ($\mu$g m$^{-2}$ s$^{-1}$) from biomass burning on 7th November 2021**

Figure 7

The last part of this comment has already been addressed in reply to comment 1 of the first reviewer. We are rewriting the relevant part of it here.

As mentioned in line number 415-421 of the original manuscript, we believe that the main reason behind the mismatch between the model and observations is due to the missing policy interventions in the model framework. We further note that the fire activity in the neighboring states of Punjab and Haryana during that period was also on a declining trend (Fig.1, Govardhan et al., 2023), so the associated fire emissions may not be completely responsible for this behavior of the model. Though we agree that the fires had not completely ceased by then, so there is a possibility of fire emissions being wrongly carried to Delhi in the model's atmosphere, unlike the observations. While carefully carrying out the analysis suggested by the reviewer, we have identified one issue with the model simulations carried out for that time period. The prescribed anthropogenic emissions for that period seem to be erroneous. They appear to have deviated from their original values, possibly due to some runtime errors. So, we have decided to redo the model simulations for those 7 days. We will correct the erroneous prescription of anthropogenic emissions and then examine how the model performs during that period. The simulations have already started. However, carrying out those in forecast mode with all the necessary data to be ingested in them is a challenging task. We are currently working on that. We will include this analysis in the modified version of the manuscript. We believe that these new simulations though, would modify the model-observation comparison for those 7 days, would not drastically impact the overall conclusions of this study. We further believe that even after correcting for emissions, though the model's performance would improve, it would still overestimate in comparison with observation owing to the absence of policy interventions in the simulations. In the revised manuscript, we will include an analysis of the factors causing the overestimation in modeled values.

P16, Table 4: Why does the POD for the "Poor". AQI category decrease by 30% to 40% in the Day 4 and Day 5? But the Accuracy doesn't change a lot. I would suggest the authors to give some thresholds for these statistical parameters indicating the confidence level and reliability of the DSS system, which may be more helpful for the policymakers. Maybe the authors can add the reference curves in Figure 6.

- The Probability of Detection (POD) comprehends the ability of the model in giving correct forecast for occurrence of an event. On the other hand, 'Accuracy', describes the ability of the model in giving correct forecast of an event or a non-event too. Thus, Accuracy encompasses the event and non-event space, while POD cover only the event space. For the 'Poor' AQI category, it may be noted that during the post-monsoonal season (fig.5b) after 27[th] October 2021, the observed AQI is always greater than 200 i.e. above the 'poor' category. Thus, as far as the 'Poor' AQI category is concerned, all those instances are recognized as 'non-events'. The model simulated AQI on most of those instances (if not all) is seen to be greater than 200, thus correctly giving forecasts of non-event. This correct forecasts of non-events mainly results in respectable value of Accuracy for the model forecasts as far as 'Poor' AQI category is concerned. On the other hand, from 27[th] October 2021 to 30[th] November 2021, the POD for 'Poor' AQI does not exist as the observed AQI does not exist in 'Poor' category. Prior to 27[th] October, the Observed AQI does exist in the 'Poor' category, the model forecasts for Day-4 and Day-5 fail to capture

that on certain occasions. This failure results in lesser POD for Day-4 and Day-5 forecast in capturing AQI in 'Poor' Category.

With reference to table 4, there are no strict guidelines in the literature about threshold values or acceptable values for the statistical evaluation parameters like Accuracy, False Alarm Ratio, Probability of Detection, Critical Success Index and Bias for air quality modeling applications. The ideal values for those parameters are 100%, 0, 100%, 100%, and 1 respectively. However, the 'acceptable' values really depend upon the criticality of the application. Thus, there is no clear indication about the acceptable values of these parameters in the literature at least for air quality forecasting applications. Nonetheless, as already mentioned in the manuscript, there exist guidelines about the acceptable values of the other statistical parameters namely the Normalized Mean Error (NME) and Normalized Mean Bias (NMB) for $PM_{2.5}$ forecasts in Emery et al., 2017, which have been extensively referred in our manuscript. According to the study, the forecasts for 24-hour-mean $PM_{2.5}$ with NMB <= 30% and NME <= 50% can be considered to be satisfactory. It can be noted from table 2 and 3, that these values for DSS for both the seasons through Day 1 to Day 5, are always lesser than the recommended thresholds. Thus, on the basis of these DSS can be considered to be doing a satisfactory performance in simulating $PM_{2.5}$ and thus the associated AQI. Owing to absence of such threshold values of Accuracy, False Alarm Ratio, Probability of Detection, Critical Success Index and Bias, we are not modifying the figure 6.

P20, Section 3.3: The two reduction scenarios show a very good linearly relationship as the authors include very little secondary PM2.5 species in the DSS system. I can understand that the authors use the GOCART aerosol scheme for computational efficiency. But it would be better to implement some previous studies on the contributions of secondary PM2.5 to total PM2.5 in Delhi.

 - We thank the reviewer for this important suggestion. We agree with the reviewer. The linearity mainly comes owing to the absence of secondary $PM_{2.5}$ species. In the next version of DSS, we are aiming to include the missing secondary aerosols by using a simple parameterization (Hodzic and Jimenez, 2011). This would include nitrate and secondary organic aerosols in the model set-up without hampering the model runtime drastically. Additionally, we are also planning to include chlorides in the model by employing a simple parameterization equation involving a relationship between Chlorides and HCl, as a function of relative humidity (Pawar et al., 2023).

**Technical corrections**

P1, Line 36: Please spell out the acronyms "NCR" when it first appears.

- The corresponding correction has been made in the revised manuscript.

P6, Line 265: Change the "SO4-2" to "SO42-" in equation (2).

- We have modified the equation in the revised version of the manuscript.

P7, Line 284 and Figure 2: Change the "SO4--" to "SO42-"

- The figure has been modified accordingly in the revised manuscript.

P20, Line 682: Change "suggests" to "suggest".

- The corresponding change has been made in the revised manuscript.

P21, Figure 8: Please make the label of x-axis clearer. The "Day-month" is a little bit confusing. I would suggest use the date for x-axis.
- We have modified the figure accordingly.

**References:**

D.H. Cusworth, L.J. Mickley, M.P. Sulprizio, T. Liu, M.E. Marlier, R.S. Defries, S.K. Guttikunda, P. Gupta, Quantifying the influence of agricultural fires in northwest India on urban air pollution in Delhi, India, Environ. Res. Lett. 13 (2018), https://doi.org/10.1088/1748-9326/aab303.

Emery, C., Liu, Z., Russell, A. G., Odman, M. T., Yarwood, G., and Kumar, N.: Recommendations on statistics and benchmarks to assess photochemical model performance, Japca. J. Air Waste Ma., 67, 582-598, https://doi.org/10.1080/10962247.2016.1265027,2017.

Gaikwad, S., Kumar, B., Yadav, P. P., Ambulkar, R., Govardhan, G., Kulkarni, S. H., ... & Ghude, S. D. (2023). Harnessing deep learning for forecasting fire-burning locations and unveiling PM 2.5 emissions. *Modeling Earth Systems and Environment*, 1-15.

Govardhan, G., Ambulkar, R., Kulkarni, S., Vishnoi, A., Yadav, P., Choudhury, B. A., ... & Ghude, S. D. (2023). Stubble-burning activities in north-western India in 2021: Contribution to air pollution in Delhi. *Heliyon*.

Hodzic, A., & Jimenez, J. L. (2011). Modeling anthropogenically controlled secondary organic aerosols in a megacity: A simplified framework for global and climate models. *Geoscientific Model Development*, *4*(4), 901-917.

Jain, S., Sharma, S.K., Vijayan, N., Mandal, T.K., 2020. Seasonal characteristics of aerosols (PM2.5 and PM10) and their source apportionment using PMF: a four year study over Delhi, India. Environ. Pollut. 262, 114337. https://doi.org/10.1016/j.envpol.2020.114337.

Jalan, I., & Dholakia, H. H. (2019). What is Polluting Delhi's Air? Understanding Uncertainties. Emissions Inventories, CSTEP, https://www.ceew.in/sites/default/files/sources-of-pollution-in-delhi-2019.pdf, last accessed on 4th September 2023

T. Liu, M.E. Marlier, R.S. DeFries, D.M. Westervelt, K.R. Xia, A.M. Fiore, L.J. Mickley, D.H. Cusworth, G. Milly, Seasonal impact of regional outdoor biomass burning on air pollution in three Indian cities: Delhi, Bengaluru, and Pune, Atmos. Environ. 172 (2018) 83–92,

https://doi.org/10.1016/j. Atmosenv.2017.10.024.

Pawar, P. V., Ghude, S. D., Govardhan, G., Acharja, P., Kulkarni, R., Kumar, R., ... & Sutton, M. A. (2023). Chloride (HC$V$ Cl−) dominates inorganic aerosol formation from ammonia in the Indo-Gangetic Plain during winter: modeling and comparison with observations. *Atmospheric Chemistry and Physics*, *23*(1), 41-59.

Sahu, S. K., Mangaraj, P., & Beig, G. (2023). Decadal growth in emission load of major air pollutants in Delhi. *Earth System Science Data Discussions*, *2023*, 1-39.

Sharma, S.K., Mandal, T.K., 2017. Chemical composition of fine mode particulate matter (PM 2.5) in an urban area of Delhi, India and its source apportionment. Urban Clim. 21, 106–122. https://doi.org/10.1016/j.uclim.2017.05.009.

Sharma, S.K., Sharma, A., Saxena, M., Choudhary, N., Masiwal, R., Mandal, T.K., Sharma, C., 2016. Chemical characterization and source apportionment of aerosol at an urban area of Central Delhi, India. Atmos. Pollut. Res. 7, 110–121. https://doi.org/10.1016/j.apr.2015.08.002.

Yadav, S., Tripathi, S. N., & Rupakheti, M. (2022). Current status of source apportionment of ambient aerosols in India. *Atmospheric Environment*, *274*, 118987.

---

## Author Response (AR2)

**Decision Support System (v 1.0) for Air Quality Management in New Delhi, India**

Gaurav Govardhan[1,2,*], Sachin D. Ghude[1,*], Rajesh Kumar[3], Sumit Sharma[4], Preeti Gunwani[5], Chinmay Jena[5], Shubhangi Ingle[1], Prafull Yadav[1], Sreyashi Debnath[1,6], Pooja Pawar[1], Prodip Acharja[1,6], Rajmal Jat[1], Gayatry Kalita[1], Rupal Ambulkar[1], Santosh Kulkarni[7], Akshara Kaginalkar[7], Vijay K Soni[5], Ravi S. Nanjundiah[8,9], and Madhavan Rajeevan[10]

1: Indian Institute of Tropical Meteorology, Ministry of Earth Sciences, Pune, Maharashtra, India
2: National Centre for Medium-Range Weather Forecasting, Ministry of Earth Sciences, Noida, Uttar Pradesh, India
3: National Center for Atmospheric Research, Boulder, CO, United States of America
4: The Energy and Resources Institute, Delhi, India
5: India Meteorology Department, Ministry of Earth Sciences, Delhi, India
6: Savitribai Phule Pune University, Pune, Maharashtra, India
7: Centre for Development of Advanced Computing, Pune, Maharashtra, India
8: Centre for Atmospheric and Oceanic Sciences, Indian Institute of Science, Bengaluru, India.
9: Divecha Centre for Climate Change, Indian Institute of Science, Bengaluru, India.
10: National Centre for Earth Science Studies, Thiruvananthapuram, Kerala, India.
* : corresponding authors:
 Gaurav Govardhan: gaurav.govardhan@tropmet.res.in ,
Sachin D. Ghude: sachinghude@tropmet.res.in

**Reply to the reviewers**

We thank the reviewers for their insightful comments on the manuscript. The comments/suggestions have certainly further improved the state of the manuscript. In this document, we have provided point by point reply to the comments/suggestion and have modified the manuscript wherever required. The reviewer's comments are written in black font, and the authors' replies are written in blue font.

**General comments**

Most of my comments have been addressed by the authors, which has improved this manuscript. I would suggest that the authors to implement some of the replies in the manuscript. I am impressed that the authors were able to correct the errors in the anthropogenic emissions and

update the results. However, I have a few questions after reviewing the revised manuscript. Please see the specific comments listed below.

**Specific comments**

P4, Line 179: The authors use the anthropogenic emission inventory from TERI for the year 2016. Is there any increasing or decreasing trends of anthropogenic emissions from 2016 to 2022?
Regarding this question, I would suggest the authors to implement key points and citations in the manuscript based on their reply, such as the increases in emissions from transport and industrial sectors over the last a few years. This can provide more information to readers.
- We thank the reviewer for this important suggestion. In the revised version of the manuscript, we have included this discussion in the main text (line numbers 186–191).

P4-5, Line 190-208: I would suggest the authors to add a figure showing the timeseries of estimated daily fire emissions in the domain during the simulation period. I'm wondering if the authors evaluated their forecasted fire emissions using other available fire emission inventories such as GFED. I'm also wondering if this climatological method could generally capture the day-to-day variability of fire emissions in this region.

- We thank the reviewer for raising this critical concern. We have examined the prescribed fire emissions in the model domain, especially over the agricultural-fire-prone region (i.e., the states of Punjab and Haryana in the northwestern part of India). As suggested by the reviewer, we have compared the total prescribed daily fire emissions over these states for the period of October–November 2021 in our modeling framework (DSS) with the corresponding daily climatological emissions prepared using the FINN (Fire INventory from NCAR) dataset for the year 2022–2018. Additionally, we compare those with the corresponding emissions from the Copernicus Atmosphere Monitoring Service (CAMS) Global Fire Assimilation System (GFAS) (Kaiser et al., 2012) for the same period. Figure 1 depicts this comparison.

[Figure]

*Figure 1: Daily emissions of Organic Carbon from fires over Punjab and Haryana states of India for the period of October 2021–November 2021 from a). FINN climatology prepared using the data from 2002–2018 (Dashed black line), b). DSS modeling framework (blue line), and c). CAMS GFAS emissions database (purple line). The units are tons of Organic Carbon. The Orange line depicts daily active fire counts over the same region retrieved by the Visible Infrared Imaging Radiometer Suite (VIIRS) instrument onboard the NASA/NOAA Suomi National Polar-orbiting Partnership (Suomi NPP) satellite. The y-axis on the left side may be used for emissions, while that on the right side may be used for fire counts.*

It may be noted from Figure 1 that the fire emissions employed in DSS differ substantially from its parent i.e., the daily mean climatological data of FINN. The primary reason for the difference is the availability of the active fire count data from VIIRS. It may clearly be seen from the plot that the daily variation in fire counts effectively decides the fire emissions in the DSS framework. It is particularly evident around 19th October and 28th October when VIIRS fire counts and the corresponding prescribed fire emissions in the DSS are zero, in spite of a non-zero FINN climatology. Thus, it can be clearly noticed that the prescribed fire emissions in DSS differ reasonably from the climatology, and they do vary on a day-to-day basis.

Upon comparing with the emissions from CAMS GFAS, it can be seen that the overall temporal variation of both datasets is roughly similar. Both of them depict zero values when fire counts are zero. Both datasets peak roughly during the same time. However, the DSS fire emissions show their primary peak towards the end of October, while that for GFAS exists in the first week of November. Similarly, the peak fire counts also are seen in the first week of November. Thus, the peak in DSS emissions of fires was reached almost a week earlier compared to GFAS and VIIRS fire count data. This behavior is likely to be linked with the FINN climatology employed in DSS. There have been a few recent studies (Jethva et al., 2019; Sembhi et al., 2020; Kant et al., 2022) that have shown that the fires in this part of the world have temporally shifted by roughly 1–2 weeks owing to a policy aimed at protecting the groundwater

in the region. However, this shift in the emissions is not reflected in the FINN climatology which is computed for the period of 2002–2018. In the next version of DSS, we are aiming to revise the FINN climatology by using more recent FINN data to prepare the base emissions. Owing to the differences in fire emissions of DSS and GFAS during the month of October, the total fire emissions of organic carbon from Punjab and Haryana states come out to be 8311.80 tons for DSS and 2538.60 tons for GFAS. Nevertheless, the total estimates agree well in the month of November, with DSS giving 12653 tons of OC and GFAS showing 10447 tons. Another difference in the estimates is that the GFAS dataset shows more variability in day-to-day emissions from fires compared to that in the DSS framework. Such high variability is not seen in the VIIRS fire counts as well. Thus, the GFAS dataset generally depicts a higher day-to-day variability.

Thus, the fire emissions employed in the DSS framework do show day-to-day variability. They are not overly driven by long-term FINN climatology. However, the peak in the absolute magnitudes of the emissions in DSS looks to reach a week earlier compared to that in GFAS, and even in VIIRS fire counts data. In the next release of DSS, we will address this issue.

This description has been included in the main text (line numbers 207–216), and Figure 1 has also been added as Figure 2 in the supplementary material.

P15, Table 2: The authors mentioned that they corrected the prescribed anthropogenic emissions and re-ran the simulations. I'm wondering why the statistics for the "Poor" and "Very poor and above" categories increase significantly during the post-monsoon season in Table 2. It seems that the prescribed anthropogenic emissions contribute to the biases.

- We thank the reviewer for raising this concern. It may be noted from Figure 5 of the original manuscript and Figure 5 of the revised manuscript that the model fails to capture the high AQI values that occurred in the first week of November 2021. As mentioned in the main text, this underestimation is related to the limitations of the modeling framework in getting the correct prescription of the emissions of particulate pollutants from stubble-burning activities occurring in the neighboring states of Punjab and Haryana. Moreover, the pulse of fire-crackers during the day of the Diwali festival would have also played a role, as mentioned in the main manuscript. However, in the original manuscript with incorrect anthropogenic emissions in the following week the model overestimated the observations substantially. This overestimation negated the underestimation in the previous weeks, resulting in the cancellation of the biases and a net low bias in the $PM_{2.5}$ concentrations. However, upon correcting the anthropogenic emissions for the week following the Diwali festival, the large overestimation in the model gets corrected. This results in a net underestimation from the model side for the post-monsoon season, as the pre-Diwali underestimation dominates the performance statistics. Thus, the correction of emissions results in increased bias. However, it may be noted that the corrections reduce the mean error values, especially for the 'very-poor and above' categories. Thus, while the biases increase due to corrections, the errors actually reduce.

**Technical corrections**

P5, Line 197-199: Please directly use the acronyms "VIIRS" and "Suomi NPP".
- We thank the reviewer for pointing this out. We have made this correction.

P14: Figure 5a: Please correct the label "Date in 2011". I think it is supposed to "Date in 2021".
- The corresponding correction has been made.

P25, Line 813: "…17.1% and 10.2…%
- The typo has been corrected.

P26, Line 816-819: Change the unit "ug/m3" to "µg m-3".
- The units have been changed wherever they appear in the entire manuscript.

P33, Line 1125: please remove the repeated citation.
- The corresponding correction has been made.

P34, Line 1155: the font for this citation looks very different.
- The font has been corrected.

**References:**

Jethva, H., Torres, O., Field, R. D., Lyapustin, A., Gautam, R., & Kayetha, V. (2019). Connecting crop productivity, residue fires, and air quality over northern India. *Scientific Reports*, *9*(1), 16594.

Kant, Y., Chauhan, P., Natwariya, A., Kannaujiya, S., & Mitra, D. (2022). Long term influence of groundwater preservation policy on stubble burning and air pollution over North-West India. *Scientific Reports*, *12*(1), 2090.

Kaiser, J. W., Heil, A., Andreae, M. O., Benedetti, A., Chubarova, N., Jones, L., Morcrette, J.-J., Razinger, M., Schultz, M. G., Suttie, M., and van der Werf, G. R. (2012). Biomass burning emissions estimated with a global fire assimilation system based on observed fire radiative power. BG, 9:527-554.

Sembhi, H., Wooster, M., Zhang, T., Sharma, S., Singh, N., Agarwal, S. Boesch, H., Gupta,, S., Misra, A., Tripathi, S.N., Mor, S., Khaiwal, R (2020). Post-monsoon air quality degradation

across Northern India: assessing the impact of policy-related shifts in timing and amount of crop residue burnt. *Environmental Research Letters*, *15*(10), 104067.

---

## Author Response (AR3)

**Decision Support System (v 1.0) for Air Quality Management in New Delhi, India**

Gaurav Govardhan[1,2,*], Sachin D. Ghude[1,*], Rajesh Kumar[3], Sumit Sharma[4], Preeti Gunwani[5], Chinmay Jena[5], Shubhangi Ingle[1], Prafull Yadav[1], Sreyashi Debnath[1,6], Pooja Pawar[1], Prodip Acharja[1,6], Rajmal Jat[1], Gayatry Kalita[1], Rupal Ambulkar[1], Santosh Kulkarni[7], Akshara Kaginalkar[7], Vijay K Soni[5], Ravi S. Nanjundiah[8,9], and Madhavan Rajeevan[10]

1: Indian Institute of Tropical Meteorology, Ministry of Earth Sciences, Pune, Maharashtra, India
2: National Centre for Medium-Range Weather Forecasting, Ministry of Earth Sciences, Noida, Uttar Pradesh, India
3: National Center for Atmospheric Research, Boulder, CO, United States of America
4: The Energy and Resources Institute, Delhi, India
5: India Meteorology Department, Ministry of Earth Sciences, Delhi, India
6: Savitribai Phule Pune University, Pune, Maharashtra, India
7: Centre for Development of Advanced Computing, Pune, Maharashtra, India
8: Centre for Atmospheric and Oceanic Sciences, Indian Institute of Science, Bengaluru, India.
9: Divecha Centre for Climate Change, Indian Institute of Science, Bengaluru, India.
10: National Centre for Earth Science Studies, Thiruvananthapuram, Kerala, India.
* : corresponding authors:
 Gaurav Govardhan: gaurav.govardhan@tropmet.res.in ,
Sachin D. Ghude: sachinghude@tropmet.res.in

**Reply to the Editor**

We thank the editor for the suggested technical correction. We have modified the "Code Availability" section and have updated the links with the inclusion of the Digital Object Identifier (DOI).